# Thermal infrared shadow-hiding in GOES-R ABI imagery: snow and forest temperature observations from the SnowEx 2020 Grand Mesa field campaign

Steven J. Pestana[1], C. Chris Chickadel[1,2], Jessica D. Lundquist[1]

[1]Civil and Environmental Engineering, University of Washington, Seattle 98105, USA
[2]Applied Physics Laboratory, University of Washington, Seattle 98105, WA, USA

*Correspondence to*: Steven J. Pestana (spestana@uw.edu)

**Abstract.** The high temporal resolution of thermal infrared imagery from the Geostationary Operational Environmental Satellites R-series (GOES-R) presents an opportunity to observe mountain snow and forest temperatures over the full diurnal cycle. However, the off-nadir views of these imagers may impact or bias temperature observations, especially when viewing a surface composed of both snow and forests. We used GOES-16 and -17 thermal infrared brightness temperature observations of a flat snow and forest-covered study site at Grand Mesa, Colorado, USA, to characterize how forest coverage and view angle impact these observations. These two geostationary satellites provided views of the study area from the southeast (134.1° azimuth, 33.5° elevation) and southwest (221.2° azimuth, 35.9° elevation), respectively. As part of the NASA SnowEx field campaign in February 2020, coincident brightness temperature observations from ground-based and airborne IR sensors were collected to compare with those from the geostationary satellites. Observations over the course of two cloud-free days spanned the entire study site. The brightness temperature observations from each dataset were compared to find their relative differences, and how those differences may have varied over time and/or as a function of varying forest cover across the study area. GOES-16 and -17 brightness temperatures were found to match the diurnal cycle and temperature range within ~1 hour and ± 3 K of ground-based observations. GOES-16 and -17 were both biased warmer than nadir-looking airborne IR and ASTER observations. The warm biases were higher at times when the sun-satellite phase angle was near its daily minimum. The phase angle, the angle between the direction of incoming solar illumination and the direction from which the satellite is viewing, reached daily minimums in the morning for GOES-16 and afternoon for GOES-17. In morning observations, warm biases in GOES-16 brightness temperature were greater for pixels that contained more forest coverage. The observations suggest that a "thermal infrared shadow-hiding" effect may be occurring, where the geostationary satellites are preferentially seeing the warmer sunlit sides of trees at different times of day. These biases are important to understand for applications using GOES-R brightness temperatures, or derived land surface temperatures (LST), over areas with surface roughness features, such as forests, that could exhibit a thermal infrared shadow-hiding effect.

# 1 Introduction

Mountain areas that receive seasonal snow are the headwaters of rivers that more than a billion people depend on globally (Immerzeel et al., 2020). Despite their importance, these are notoriously difficult places to gather hydrological or meteorological observations for predicting snow water equivalent (SWE) and the timing and magnitude of streamflow (Raleigh et al., 2013). Longwave radiation measurements, of which the upwards component is controlled by the diurnal cycle of snow surface temperature, have been identified as especially critical for modeling these snowmelt fed systems (Lapo et al., 2015;

Raleigh et al., 2016). Distributed observations of surface temperatures at sub-daily temporal resolutions are needed for hydrologic and land surface models, and could aid real-time forecasting (Shamir and Georgakakos, 2014). Thermal infrared imagery from geostationary satellites that constantly view the same portions of Earth's surface, such as GOES-R Advanced Baseline Imager (ABI), can make land surface temperature (LST) observations at very high temporal resolution (5 minutes or better), capturing the full diurnal cycle. These observations, however, have spatial resolutions of 2+ km, and view the land

surface from off-nadir angles.

    The 2020 NASA SnowEx field campaign was a collaborative effort between government agencies and academic researchers to intercompare and evaluate snow remote sensing methods with extensive ground-based observations. This was conducted in early 2020 at Grand Mesa, a large flat-topped mountain in the western part of the US state of Colorado. As part of this campaign, a multi-sensor experiment was designed to investigate how the off-nadir views of GOES-R satellites affect their

surface temperature retrievals over snow and forests by making thermal infrared brightness temperature observations and intercomparisons at a range of spatial and temporal scales (Table 1, Figure 1). This unique study site, a flat expanse of snow and conifer forest, allowed us to investigate how forests affect observed brightness temperatures, independent of the effects due to complex terrain. Ground based snow brightness temperature measurements provided a continuous point of comparison for GOES-R, while multiple overpasses from airborne IR imagery, gridded to 5 m spatial resolution, provided finer resolution

distributed brightness temperature details over the course of two mornings. To benchmark the ground point measurements and airborne IR, which itself has a wide range of view angles (Pestana et al., 2019), we compared these with a coincident nadir-looking ASTER thermal infrared image at 90 m spatial resolution. Specifically, we set out to address the following questions regarding GOES-R ABI thermal infrared brightness temperature observations during SnowEx: 1) What were the relative accuracies of each source of remotely sensed brightness temperature? 2) How did fractional forest cover impact the relative

accuracy of GOES-R ABI brightness temperature across the study area? 3) How did the relative accuracy of GOES-R ABI brightness temperature change over the course of each day of observations?

    We hypothesized that among the brightness temperature observations collected, the best agreements would be between the nadir-looking ASTER, nadir-looking Airborne IR, and ground-based snow brightness temperatures. We further hypothesized that for GOES-R ABI pixels with greater forest canopy, the observed brightness temperatures would be greater than those

from the nadir ASTER and airborne IR imagery. Finally, we hypothesized that these warm biases would be greatest in the

early morning observations by GOES-16 (East) and late afternoon observations by GOES-17 (West), when they are viewing the solar-illuminated side of trees.

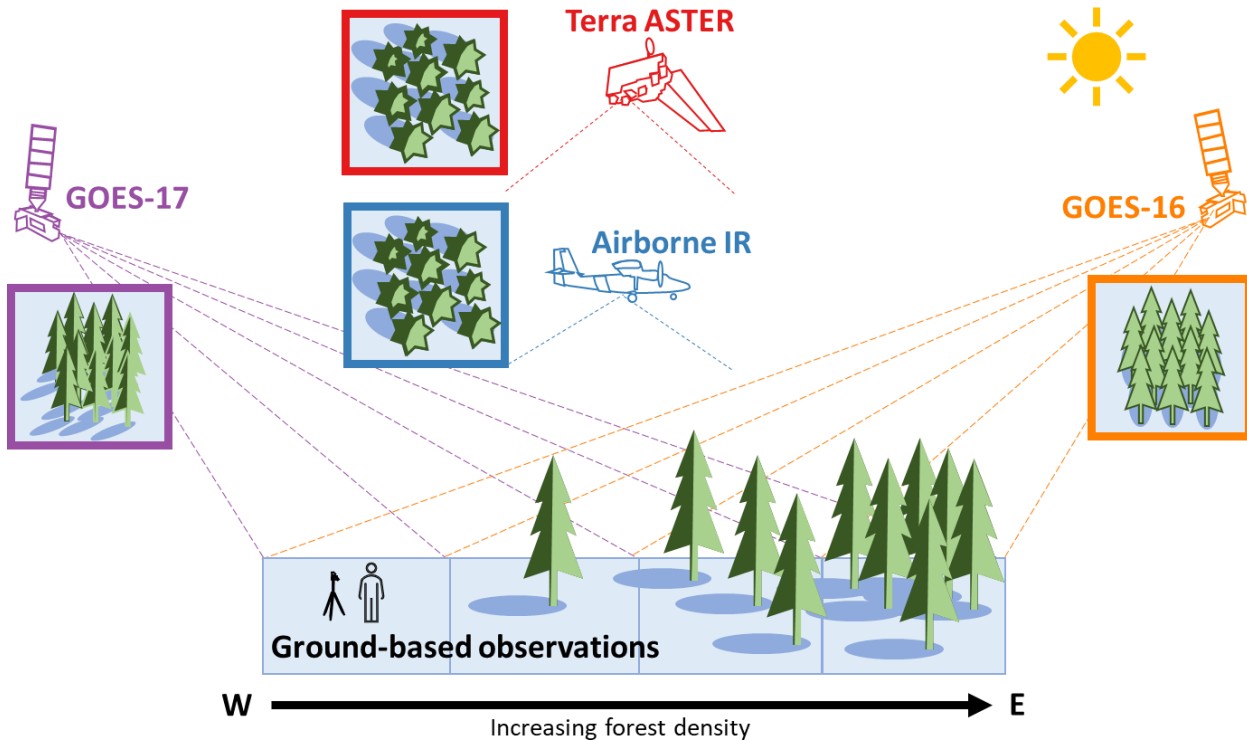

**Figure 1. Conceptual illustration of how nadir and off-nadir looking remote sensing imagers see parts of a forest canopy and, depending on the direction of solar illumination, the shadows cast by trees.**

## 2. Background

### 2.1 High temporal resolution thermal infrared imagery

Land surface models are highly sensitive to their temperature and longwave forcing input (Mizukami et al., 2014; Raleigh et al., 2015), in both accumulation and ablation periods (Günther et al., 2019). This is especially important for sparsely instrumented mountain areas, where land surface models can have air temperature errors of ~3-4 °C (Tomasi et al., 2017). Differences in forcing inputs, or how surface energy fluxes are parameterized in land surface models, can result in hourly surface temperature errors of as much as 15 °C (Essery et al., 2013), and lead to snow disappearance date uncertainties spanning months (Hinkelman et al., 2015). Surface temperature observations at model-relevant time steps, such as hourly temporal resolutions, are needed especially to capture diurnal processes like snow melt-freeze cycles (Niu et al., 2011) and snow grain metamorphism, which in turn drive feedbacks in the surface energy balance through changes in emissivity and albedo (Flanner and Zender, 2006; Warren, 1982, 2019).

Thermal infrared satellite imagery can provide snow surface temperature observations for homogenous snow-covered landscapes (Hall et al., 2008; Wan et al., 2002), estimates of near-surface air temperature (Pepin et al., 2016; Shamir and Georgakakos, 2014), and dewpoint temperature (Raleigh et al., 2013), all of which are needed for modeling hydrologic

processes. However, satellite observations at <100 m spatial resolution are made too infrequently (4-16 day repeat) for looking at snow surface energy balance processes at model-relevant timesteps. Observations from imagers like VIIRS or MODIS (250 m to 1 km resolution) provide two observations per day each for mid-latitude locations. The observations from the sun-synchronous orbiting MODIS or VIIRS imagers do not necessarily see the coldest and warmest times of day to capture the full diurnal temperature range (DTR), nor do they provide LST more frequently than every several hours. Their twice daily

observations can also be obscured by cloud cover, creating large data gaps relative to the diurnal cycle of snow surface temperatures.

Geostationary satellite imagery may help overcome these drawbacks, providing high temporal resolution LST, potentially seeing between intermittent periods of cloud cover, though at coarser spatial resolutions (2+ km) and off-nadir view angles. In the complex terrain and forest vegetation of mountain watersheds, the individual image pixels from thermal infrared

observations will report an LST signature that is a mixture of the subpixel snow and forest surface temperatures (Dozier, 1981; Selkowitz et al., 2014). Snow and vegetation can have significant temperature differences, especially on clear days where incoming solar radiation warms forest canopies more than the high albedo snow surface and during the snowmelt period, when daytime snow surface temperatures are capped at 0 °C (Pestana et al., 2019). Sections 4.4 and 5.4 describe how we tested the uncertainty around the assumption that the brightness temperatures of surfaces within a pixel's footprint scale linearly to a

mean brightness temperature, and the geolocation uncertainty of GOES ABI pixel footprints.

## 2.2 Off-nadir views and shadow-hiding

Imagery from geostationary satellites comes with the drawback of having off-nadir view angles. These view angles are dependent on the location of the area of interest on Earth's surface and the satellite's orbital position (Schmit et al., 2017). Interpretation of off-nadir thermal infrared images require consideration of the parallax effect over rough surfaces, the angular

emissivity of different surface materials, and longer atmospheric path lengths.

In thermal infrared satellite imagery, the parallax effect can lead to different brightness temperatures being observed for the same area of interest at different view angles. Geostationary satellites view the mountains of North America from the south; therefore, south-facing mountain slopes appear lengthened, occupying a larger portion of an image, north-facing mountain slopes are foreshortened, and steeper north-facing mountain slopes may be completely occluded from view. Prior work

compared brightness temperatures from off-nadir GOES-16 thermal infrared imagery to coincident nadir-looking ASTER and MODIS thermal infrared imagery over the Sierra Nevada of California (Pestana and Lundquist, 2022). This work showed that GOES-16 imagery preferentially viewed south-facing slopes, which receive more solar illumination in the daytime, heating up more than shaded north-facing slopes. GOES-16 brightness temperatures were therefore biased warm in comparison to those from ASTER and MODIS, which being nadir-pointing, could see both the sunlit and shaded sides of mountain slopes.

The parallax effect is also important at smaller spatial scales, such as that of individual trees in a forested landscape (Figure 1). Tree profiles (rather than canopy tops) come into view when observed from off-nadir angles, causing each tree to take up more space in the image. Off-nadir imagery of landscapes with snow and forests will also have trees obscuring the snow surface behind and beneath them (Balick et al., 2002; Pestana et al., 2019). Much like at the scale of mountain slopes, daytime solar illumination, especially at low sun angles, will warm up one side of trees more than the other shaded side. The observed

thermal infrared brightness temperature of mountainous or forested terrain is therefore dependent on both view angle and the angle of solar illumination. The angle between the two is the phase angle (Henderson et al., 2003). The same landscape can therefore appear warmer at small phase angles when solar illuminated surfaces are in view and shaded areas are occluded, or colder at large phase angles when more shaded areas are in view and sunlit areas are occluded (Figure 1). With visible and NIR imagery, this effect is referred to as "shadow-hiding," or creating a "hotspot" (really a bright spot), over surface roughness

features, such as trees (Hall et al., 1993; Bréon et al., 2002) or wind forms in snow, such as sastrugi (Warren et al., 1998). With nadir-pointing satellite imagers in sun-synchronous orbits (e.g. MODIS or VIIRS), the phase angle changes slowly on an annual cycle, but with a geostationary satellite able to observe at high temporal resolution, the phase angle changes continuously over each diurnal cycle (Pestana et al., 2023).

Due to the off-nadir view angles of the GOES-R ABI observations, the angular emissivity of surfaces observed, a snow-

covered forested landscape, must also be considered. The near blackbody-like emissivity of vegetation, such as the conifers that dominate the study area (Section 3.1), does not vary with view angle. The emissivity of snow, however, does vary with view angle and with grain size (Dozier and Warren, 1982). The emissivity of snow is smaller at larger view angles, however over rough snow surfaces such as the wind-formed sastrugi observed over the westernmost portion of the mesa, there is a wide distribution of view angles normal to the snow surface even for a nadir-looking imager. At the view angles that GOES-16 and

-17 observed our study area (Section 3.3.1), snow emissivity could range from 0.95 for coarse-grained snow (such as > 1 mm melt forms) to 0.99 for fine-grained snow (such as < 1 mm fragmented dendritic precipitation particles) (Hori et al., 2006; Warren, 2019). At the lowest end of this emissivity range, a snow surface at 260 K would have a brightness temperature about 3 K colder seen by the off-nadir looking GOES compared to a nadir looking satellite imager, and about 0.5 K colder at the highest end of this range. Snow pit observations (Vuyovich et al., 2021) coinciding with the GOES observations used in this

work reported predominately fine-grained decomposing and fragmented precipitation particles of < 1 mm at the snow surface. We can therefore expect to see in our comparison of brightness temperatures from imagers with different view angles that GOES ABI observations may be biased low by 0.5 K relative to nadir observations due to emissivity alone. GOES-R ABI brightness temperatures may also be biased low in comparison with a nadir-looking view due to their longer atmospheric path lengths, but this effect can be negligible for cloud-free high-altitude winter conditions when absorption of thermal infrared

radiation by water vapor is minimal (Pestana and Lundquist, 2022). See Section 6.1 for further discussion of atmospheric effects.

# 3 Study site and observations

## 3.1 SnowEx 2020 field campaign study site

The 2020 NASA SnowEx field campaign intensive observation period (IOP) took place at Grand Mesa in western Colorado (39.02°, -108.12°) from 26 January to 14 February 2020 (Figure 2). This period of the field campaign brought together snow remote sensing researchers to test new instruments and methods, and to collect extensive ground-based observations for validation. Grand Mesa, a large flat-topped mountain with elevations above 3000 m, is located within the watersheds of the upper Colorado river and its tributary, the Gunnison River. This region was chosen as a location to evaluate remote sensing observations of mountain snow because of its flat terrain, where the additional complications of view angles and complex terrain are minimized. The site is also beneficial for thermal infrared remote sensing because at its high elevation, the atmospheric path length, and therefore magnitude of absorption of thermal infrared radiation by water vapor in the atmosphere, is lower than that of sites at lower elevations. The high emissivities of both snow and conifer trees provide us with a scene where brightness temperatures are close to true surface temperatures (Kim et al., 2018; Warren, 2019).

During the IOP field campaign, the ground surface was entirely snow-covered, with no bare ground surfaces visible in remotely sensed imagery. The westernmost portion of Grand Mesa is sparsely forested, and forest cover increases across the mesa towards the east. Mixed conifer forests of Engelmann spruce (*Picea engelmannii*), subalpine fir (*Abies lasiocarpa*), and lodgepole pine *(Pinus contorta* var. *latifolia*) species dominates the vegetation that stood above the snow, with some stands of deciduous Aspen (*Populus tremuloides*) trees (Currier et al., 2019).

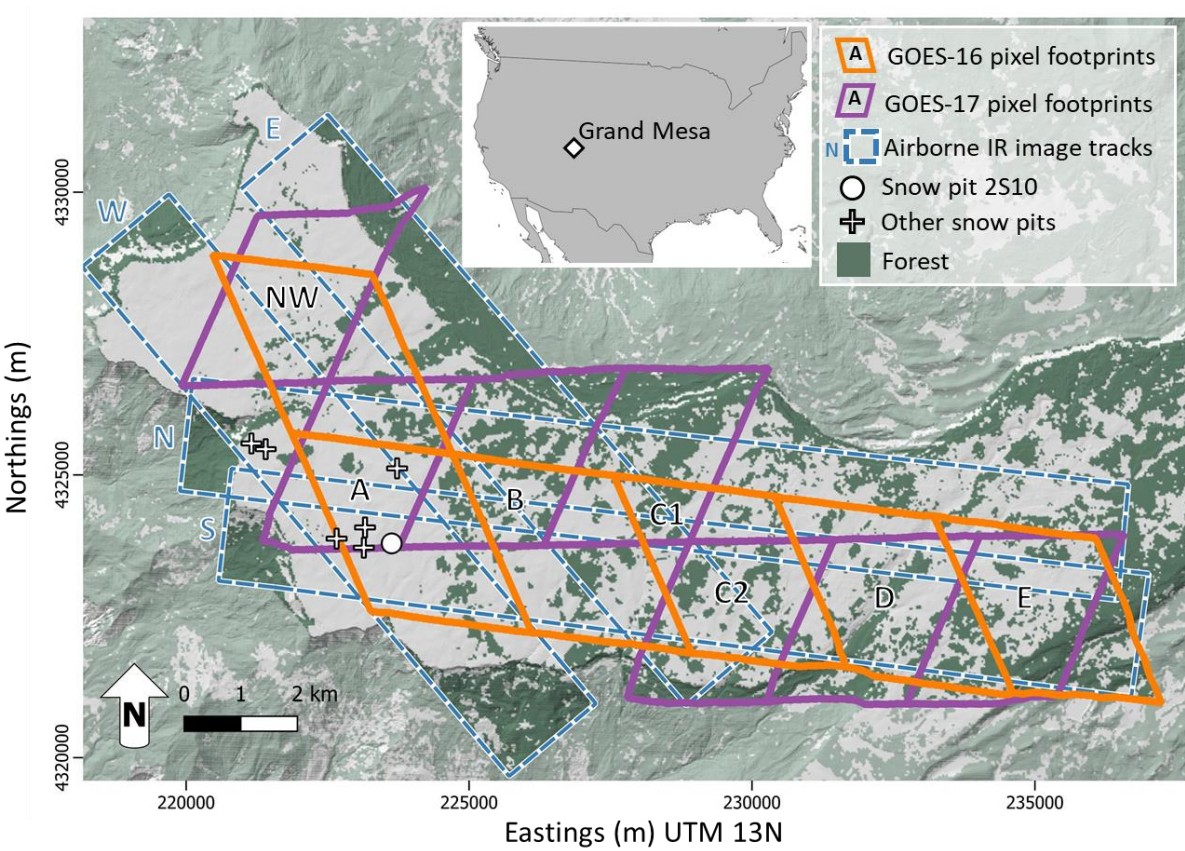

**Figure 2. Map of the study area at Grand Mesa, Colorado, and inset map showing its location within the contiguous United States. Polygons outline GOES-16 (orange) and GOES-17 (purple) ABI pixel footprints, and airborne IR image mosaic swaths (blue dashed lines). Snow pit #2S10, where automated continuous snow brightness temperatures were observed, is indicated by the white circle, and instantaneous snow surface temperature observations at other snow pits are indicated by white +'s. Dark green areas indicate forests within the area covered by the thermal infrared remote sensing imagery.**

## 3.2 Ground-based observations

Ground-based observations at Grand Mesa included continuous automated measurements of snow brightness temperature, and instantaneous manual snow surface temperatures measurements taken as part of the data collection at individual snow pits. Snow brightness temperatures were measured continuously by an Apogee SI-111 radiometer (8 – 14 μm) installed at snow pit #2S10 in the western portion of the mesa (39.0195, -108.19214). This radiometer viewed the snow surface at an angle of 45° from nadir and was mounted 2 m above the ground surface, which at this time was 1.27 m above the snow surface. The radiometer had an instantaneous field of view of 44°, giving it an approximately elliptical footprint of 1.45 m x 2.45 m on the top of the snow surface. Snow brightness temperatures measured by this radiometer were recorded at a 5-minute temporal resolution (Pestana and Lundquist, 2021). More than 150 snow pits were dug by the field teams over the course of the IOP (Vuyovich et al., 2021), and among the measurements recorded at each snow pit were snow grain size and types, snow surface temperature, and the time of the surface temperature measurement. Snow surface temperatures were measured by a stem

thermometer inserted into the top-most 1 cm of snow and shaded from direct sunlight. These snow pit data were accessed from the SnowEx database through the snowexsql Python library (Johnson et al., 2023). The USGS National Elevation Dataset 1 arc-second (~ 30 m) DEM (US Geologic Survey, 2017) and Tree Canopy Cover (TCC) product from the National Land Cover Database (NLCD) 2016 (Coulston et al., 2012) were used to compute zonal statistics of elevation and fractional vegetated area

($f_{veg}$), respectively, across the study site.

**Table 1. Ground-based and remotely sensed temperature observations from the SnowEx 2020 field campaign used in this study.**

| Dataset | Spatial resolution | Temporal resolution | Spectral range | Specified accuracy |
|---|---|---|---|---|
| **Ground-based observations:** | | | | |
| Continuous snow brightness temperature | 1.5x2.5 m spot size | 5 minute | 8 – 14 μm | ± 0.2 K |
| Instantaneous snow surface temperature | n/a | n/a | n/a | ± 1 K |
| **Remote sensing observations:** | | | | |
| Airborne IR image mosaics | 5 m | ~10 minute | 8 – 14 μm | ± 2 K |
| ASTER IR image (AST L1T) | 90 m | n/a (one image) | 10.95 - 11.65 μm | 2% (±1.1 K @ 260 K) |
| GOES-16 and -17 ABI band 13 (ABI-L1b-RadC) | ~ 2 km | 5 minute | 10.05 – 10.55 μm | 1.5% (±0.7 K @ 260 K) |
| GOES-16 and -17 ABI band 14 (ABI-L1b-RadC) | ~ 2 km | 5 minute | 10.8 – 11.6 μm | 1.5% (±0.8 K @ 260 K) |
| GOES-16 and -17 ABI LST (ABI-L2-LSTC) | ~ 2 km | 1 hour | n/a | ± 2.5 K |

### 3.3 Remote sensing observations

### 3.3.1 GOES-R ABI

Images from the Advanced Baseline Imager (ABI) onboard GOES-16 and GOES-17 were retrieved for the duration of the study period in February 2020. The 5-minute temporal resolution Level 1b top-of-atmosphere Radiance CONUS product (L1b-RadC) for thermal infrared bands 13 (10.3 μm) and 14 (11.2 μm), and the 1-hour temporal resolution Level 2 Land Surface Temperature CONUS product (L2-LSTC), were downloaded as NetCDF files via the goespy library (Mello and Pestana, 2022). Both satellites viewed the Grand Mesa study site from similar view angles, though with GOES-16 in the southeastern sky

(azimuth 134.1°, or 45.9° from due south) and GOES-17 in the southwestern sky (azimuth 221.2°, or 41.2° from due south), with elevation of angles of 33.5° and 35.9° respectively.

The specific ABI pixel footprints that overlapped the study area on top of Grand Mesa were identified by first orthorectifying (Pestana et al., 2022; Pestana and Lundquist, 2022) 2 km L1b-RadC imagery clipped to the region surrounding Grand Mesa from each of GOES-16 and GOES-17. Vector polygons outlining the ABI pixel footprints were created from these sample

images, and the resulting polygons were then used to compute land surface elevation summary statistics from the 30 m

resolution DEM (US Geologic Survey, 2017). We sampled ABI pixels with footprints on top of Grand Mesa that covered an area with a mean elevation greater than or equal to 3000 m and standard deviation of elevation less than or equal to 60 m. For GOES-16, this resulted in six pixels selected, and for GOES-17, seven pixels (Figure 2). The pairs of overlapping GOES-16 and -17 pixels were labeled "NW" for the pixels covering the northwestern most portion of Grand Mesa, and "A" – "E" for

the pixels running roughly west to east across the study area. Two GOES-17 pixels are labeled "C1" and "C2" to indicate that they both primarily overlapped with GOES-16 pixel "C." Timeseries of the thermal infrared radiance, brightness temperatures (both from L1b-RadC), and LST (from L2-LSTC) were compiled for each of these pixels from imagery covering 8 to 15 February 2020.

These pixel footprints were used to delineate areas of different fractional vegetation cover for comparison across the mesa.

The NLCD TCC map was converted to a binary forest map with a vegetation threshold at 20% TCC. This threshold was chosen to visually match the forest above the snow surface in the ASTER visible image from the morning of 8 February (Figure 5a). For each GOES ABI pixel footprint, the fractional vegetation area ($f_{veg}$) was calculated as the fraction of the pixel footprint classified as forest in the binary forest map.

This work uses brightness temperatures computed from high temporal resolution GOES-R ABI top-of-atmosphere radiance

(L1b-RadC), rather than the LST product (L2-LSTC). Though the LST product corrects for atmospheric absorption and surface emissivity, they are only generated hourly, rather than the much higher 5-minute temporal resolution available from the radiance product. The hourly LST observations were also not available for most of the daytime periods. The ABI Cloud Mask algorithm is generally used to determine when and where the land surface is not obscured by clouds to determine if LST should be computed. However, identifying cloud cover over snow is notoriously difficult due to their similar appearance in remote

sensed imagery across the visible through infrared spectrum (Rittger et al., 2019; Stillinger et al., 2019). Only four daytime LST observations on 8 February were available, and on 11 February there were three daytime and nine nighttime LST observations. Additionally, the ground-based radiometer and airborne thermal infrared imagery (discussed in the next section) provided brightness temperature observations and were not corrected for atmospheric absorption or surface emissivity to derive an LST product from each. Therefore, we focus primarily on radiance and brightness temperature in our analysis.

**3.3.2 Airborne IR imagery**

Airborne IR imagery (Chickadel et al., 2022) was collected on four days with the UW Applied Physics Laboratory's Compact Airborne System for Imaging the Environment (CASIE), consisting of thermal infrared cameras and an infrared radiometer, mounted on the Twin Otter research aircraft from the Naval Postgraduate School (NPS) Center for Interdisciplinary Remotely Piloted Aircraft Studies (CIRPAS). CASIE was installed on the aircraft to be primarily nadir-looking, and had three DRS

UC640-17 thermal infrared cameras (8 – 14 μm) pointing with bore-sight incidence angles of 19°, 0° (nadir-looking), and 21° from port to starboard on the aircraft. These three cameras have overlapping fields of view of 25° (left camera) and 40° (center and right cameras) perpendicular to the aircraft flight direction, with a total field of view of about 72.5° (Lundquist et al., 2018). The aircraft flew at about 1 km above the top of Grand Mesa, giving the three cameras a total swath width of about 2.5

km perpendicular to the direction of flight, and a raw ground resolution of 1 m. A nadir-looking Heitronics KT15.85D infrared
radiometer with spectral range 9.6-11.5 um and narrow 1.9° field of view provides a precise brightness temperature measurement for a spot on the ground surface at the center of the center camera's field of view. This higher precision radiometer data was used to calibrate the thermal infrared cameras (Pestana et al., 2019) prior to mosaicking images together using the aircraft INS-GPS navigation data from their original ~1.1 m spatial resolution to 5 m.

Imagery from two flights on 8 February (from about 08:00 – 10:00 and 11:00 – 13:00) and one flight on 11 February (from
about 10:00 – 13:00) were used as these days had the least cloud-cover over the study site. Flightlines over the study site were along two sets of parallel tracks that would overlap with ground observations at snow pits (Figure 2). One set of parallel tracks ran east-west, and the other tracks ran roughly northwest-southeast to capture the northwest portion of the mesa. The airborne IR imagery collection was in part planned to coincide with the collection of satellite imagery by ASTER on 8 February.

### 3.3.3 Terra ASTER

The NASA Terra satellite made an overpass of the Grand Mesa study site and imaged it with ASTER at 11:07 (local time, UTC-7) on 8 February 2020. ASTER provides a reliable source of surface brightness temperature information at 90 m spatial resolution (Abrams, 2000), fine enough to capture the surface temperature variabilities across the Grand Mesa study area and resolve forest stands from open snow. For this single observation of the Grand Mesa study site, the ASTER Level 1 Precision Terrain Corrected Registered At-Sensor Radiance (AST L1T) product (Meyer et al., 2015) for band 14 (11.3 μm) was used.
The top-of-atmosphere radiance was converted to brightness temperatures (Thome, 1999) for comparison with the other ground-based and remote sensing observations.

### 4 Methods

### 4.1 Evaluating airborne IR image mosaics against ASTER

To first assess the accuracy of the airborne IR imagery, two airborne IR mosaics from 8 February at 11:07 and 11:19, running
east-west across the mesa, were compared against the coincident ASTER image captured at 11:07. The airborne IR mosaics were first resampled to the same spatial resolution of ASTER by taking the mean of the original 5 m spatial resolution images within each 90 m ASTER pixel. The differences between ASTER and each of the two resampled airborne IR mosaics were then computed, producing two difference maps, and the mean and standard deviation of differences were computed for each. Means and standard deviations of differences were also computed for the portions of the difference maps within each of the
GOES-16 ABI pixel footprints. The difference maps were inspected qualitatively for patterns in the imagery across the study site to better characterize properties of the airborne IR imagery.

## 4.2 Comparison of airborne IR, ASTER and ground observations

To determine how representative the ground-based point brightness temperature measurements were of their surrounding areas, airborne IR imagery and the single ASTER satellite image were compared with ground-based data at the times when each passed over the study site. From each airborne IR mosaic, a 1 x 1 km square was extracted from around the automated snow brightness temperature measurement site at snow pit #2S10 (Figure 3). Only airborne images that covered at least 30% of this 1 km$^2$ area were used. The mean, median, and standard deviation of brightness temperatures from this area in each airborne IR image were computed. This provided us with a timeseries of snow brightness temperatures at each time that the aircraft flew over the ground site. The same 1 km$^2$ region around the snow brightness temperature measurement site was sampled from the single ASTER image from the morning of 8 February to compute the brightness temperature mean, median, and standard deviation as seen by ASTER. The correlation between the timeseries of airborne IR snow brightness temperature observations and ground-based snow brightness temperatures were computed for each day, while the difference between the ground-based snow brightness temperature measurements and ASTER observations were computed at the time of its overpass.

To compare the snow surface temperature observations taken at each snow pit against coincident airborne IR imagery, all of the snow pits sampled from 8 February and 11 February that were along the aircraft's flight path within +/- 30 minutes of the flight overpass were compared to the images from that flight. The mean and standard deviation of airborne IR observed brightness temperatures within a 100 m$^2$ square centered on the snow pit were then compared with these ground-based observations to determine how the differences between the two varied over time and across the study area. One snow pit overflown by the aircraft was about 50 m south of a forest stand, while all other snow pits were 250 m or more away from the nearest forest stand.

The sensitivity of the comparisons between airborne IR imagery, ASTER, and point ground-based temperature observations was tested by reducing the size of the square area from which temperatures were sampled from the airborne IR images (Figure 3), from a square with sides ranging from 1000 m to 100 m (a single ASTER pixel is 90 m), and then for the airborne imagery ranging from 1000 m to 5 m (the size of a single airborne IR pixel) (Table 2). All airborne IR images were included in this analysis, rather than excluding images that covered less than 30% of the area, as was done in the prior analysis of airborne IR and ground data. The mean, median, and standard deviation of brightness temperatures were computed for the sampled region in each image. The mean difference, and root mean squared difference between all the airborne IR brightness temperature observations of these areas and the coincident ground-based temperature measurements, were computed. The smallest area sampled was a single 5 m airborne IR pixel that should contain the ~2.5 m footprint of the ground-based radiometer that was measuring snow brightness temperatures. However, the geolocation accuracy of the airborne IR mosaic imagery is only about +/- 10 m (Pestana et al., 2019). The single-pixel sampled therefore may not actually overlap completely with the ground-based radiometer footprint, but rather be directly adjacent to it.

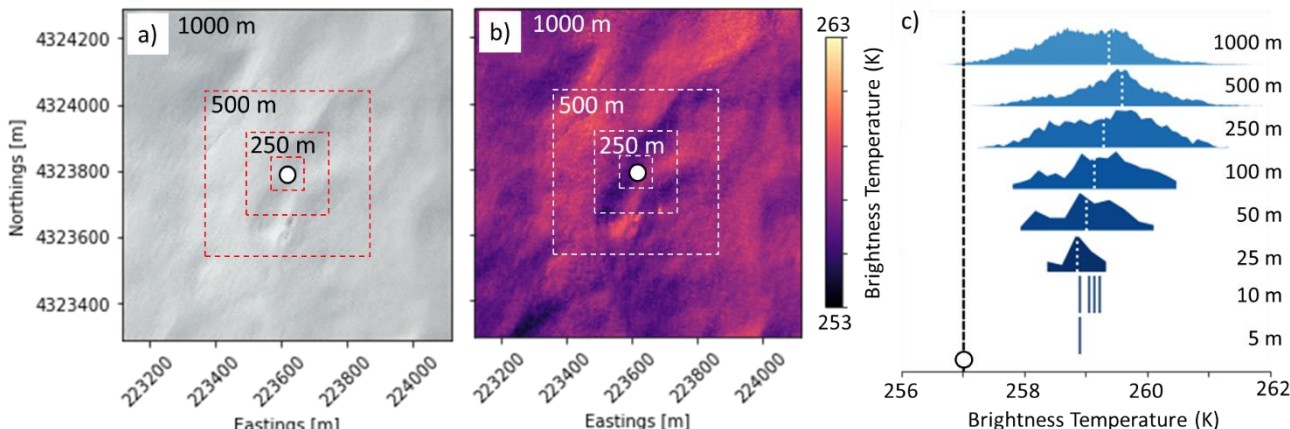

**Figure 3. Airborne a) visible, and b) IR image of the area around the snow brightness temperature observation site at snow pit #2S10. Boxes indicate regions from which the mean airborne IR brightness temperature information was taken for comparison with the ground-based observations (only the boxes with sides of 1000, 500, 250, and 100 m are shown). Map coordinates are in UTM zone 13N. c) Histograms of the airborne IR brightness temperatures from this example image plotted alongside the ground-based snow brightness temperature at this time (vertical dashed line).**

## 4.3 Comparison of high temporal resolution GOES-R ABI with continuous ground observations

GOES-16 and -17 brightness temperature observations were compared against the ground-based snow brightness temperature observations at snow pit #2S10. A timeseries of brightness temperatures for bands 13 and 14 at 5-minute temporal resolution was created for 8 to 12 February 2020 for the GOES-16 and -17 ABI pixels, which contained snow pit #2S10 (both labeled pixel A). Two cloud-free periods, 8 February (7:00 – 18:00), and 11 February (00:00 – 18:00) were manually identified by inspecting the GOES imagery and brightness temperatures for cold cloud tops obscuring the study site. The ground-based snow brightness temperature observations at snow pit #2S10 over these same time periods were resampled to match the 5-minute temporal resolution of GOES ABI brightness temperatures. All timeseries were then smoothed with a 30-minute running mean to remove the highest frequency variability (median $< \pm 0.02$, $\sigma < 0.6$ K) from the data and fill data gaps. The daily maximum and minimum temperatures and diurnal temperature range (DTR) were then found for both the ground-based snow brightness temperature observations, and for the GOES ABI brightness temperatures. The mean and root mean squared difference between GOES ABI brightness temperatures and the ground-based snow brightness temperatures were also computed. For 8 February, because there was cloud-cover at night obscuring the study area until 7:00, we only compared the timing of maximum daytime temperature between GOES and the ground-based observations.

## 4.4 Comparison of GOES-R ABI, airborne IR, and ASTER imagery

The differences between GOES-16 and -17 ABI brightness temperatures for 8 February (7:00-18:00) and 11 February (21:00 10 Feb. -18:00 11 Feb.) were computed for each pair of corresponding pixels (NW, A, B, C/C1, C/C2, D, and E) across the mesa. This comparison was performed with both ABI bands 14 and 13. The mean difference, standard deviation of differences, and range of differences for each pixel were plotted against the corresponding pixel's $f_{veg}$ value to inspect for any apparent

correlation between these differences and the forest fraction within each pixel footprint. The comparison of GOES-16 to -17 is complicated because they view the scene from different perspectives. For example, the pair of pixels "A" from GOES-16 and -17 overlap each other, but they do not have the same footprint on the ground, have slightly different values of $f_{veg}$, and may include different amounts of the edges of the mesa.

From each airborne IR mosaic, if the airborne imagery covered at least 30% of each of the GOES-16 and -17 ABI pixel footprints on top of the mesa, the region was sampled from the mosaic (Figure 4). The mean, median and standard deviation of temperatures within each footprint were computed for comparison against the ABI band 14 brightness temperature and LST of that pixel for a 10-minute window around the aircraft overpass time. The use of the GOES ABI pixel footprints to sample the finer spatial resolution airborne images assumes a direct linear scaling of the finer spatial resolution brightness temperatures to a mean brightness temperature at the coarser GOES ABI spatial resolution (Pestana and Lundquist, 2022). The sensitivity of the results to this assumption, and to the geolocation accuracy of a pixel footprint, were tested by comparing GOES ABI brightness temperatures with airborne brightness temperatures sampled from an area larger than a single pixel footprint. An example of the 500 m buffer around a GOES ABI pixel footprint is illustrated in Figure 4b. Similarly, the GOES-16 and -17 ABI pixel footprints and expanded footprints were used to extract the mean, median and standard deviation of top-of-atmosphere radiance from the single ASTER image. These were  then converted to brightness temperature for comparison against ABI band 14 brightness temperatures.

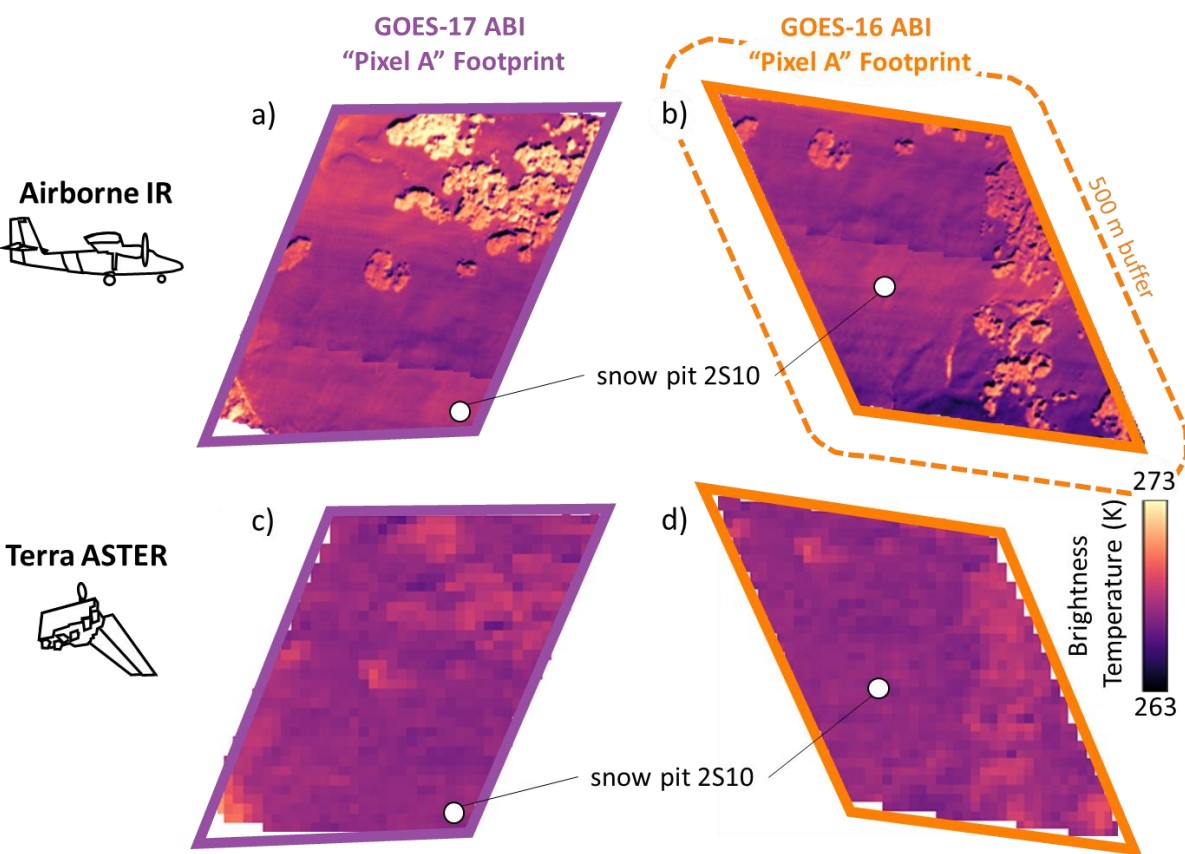

**Figure 4. Example of sampling (a,b) 5 m spatial resolution airborne IR image mosaics and (c,d) 90 m spatial resolution ASTER image using the GOES-R ABI pixel footprints, or (dashed line in b) pixel footprints with an additional 500 m buffer.**

## 5 Results

### 5.1 Evaluating airborne IR image mosaics against ASTER

The airborne IR imagery was found to have a warm bias compared with ASTER brightness temperatures. The mean differences
between the two resampled airborne IR image mosaics and the ASTER image from the morning of 8 February were 0.4 K and 0.8 K and had standard deviations of 1.5 K and 1.4 K, respectively. Using the GOES-16 ABI pixel footprints labeled A-E, we found that the mean differences between the airborne IR and ASTER did not vary with vegetation cover.

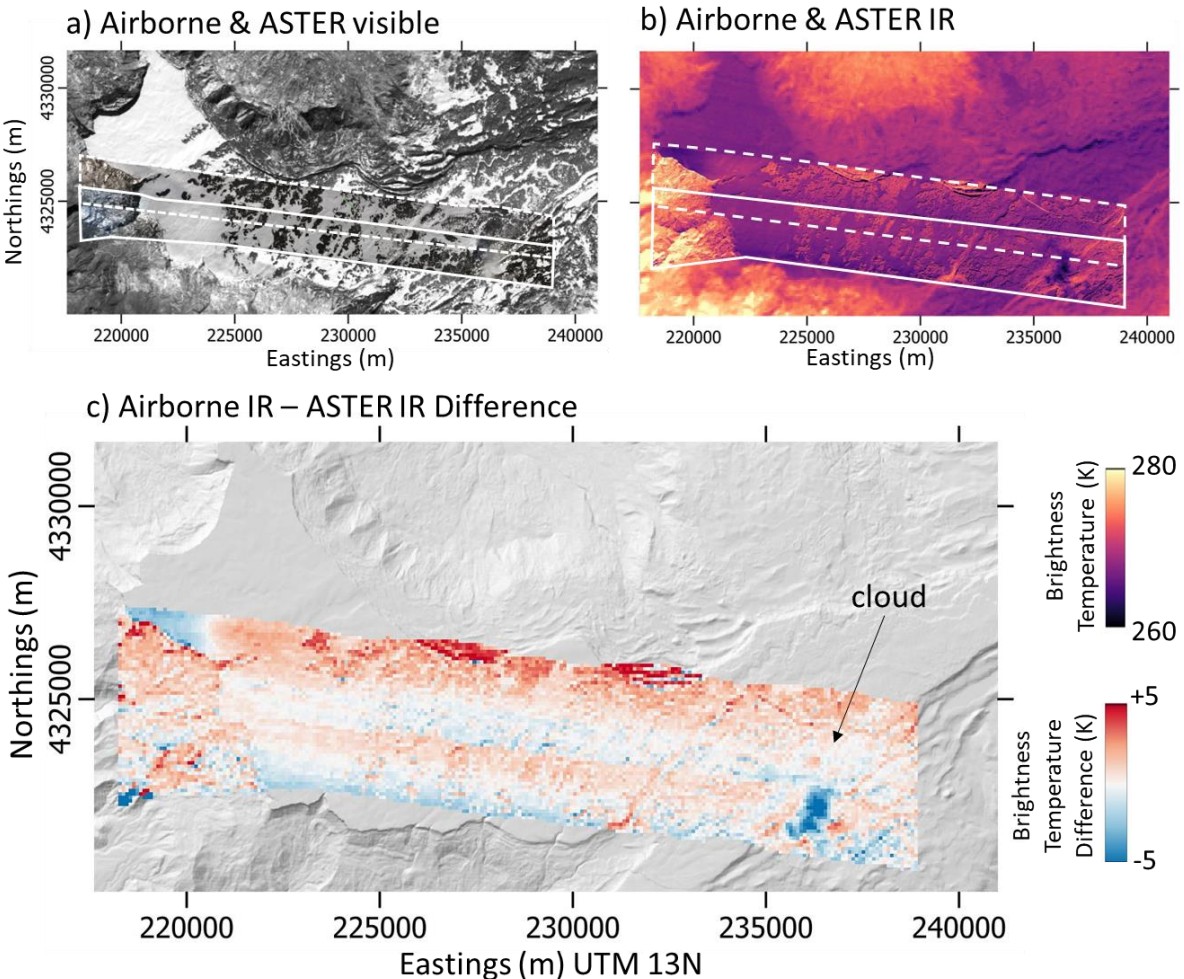

**Figure 5. Comparison of airborne and ASTER a) visible and b) IR observations on the morning of 8 February 2020. Two overlapping airborne swaths of visible (a) and IR (b) imagery are outlined in white dashed (northern flightline at 11:07 UTC-7, east to west flight direction) and solid (southern flightline at 11:19 UTC-7, west to east flight direction) lines. c) The difference between airborne IR image mosaics and the ASTER image (at 11:07 UTC-7).**

There were, however, systematic patterns in the difference between airborne IR and ASTER observed brightness temperatures. The brightness temperature difference maps for each of the two coincident airborne IR image mosaics exhibit a gradient across their field of view, perpendicular to the aircraft's flight direction (Figure 5). Along the southern edge of each image mosaic, the airborne IR brightness temperatures are about 0.5 K colder than ASTER brightness temperatures, and along the northern edge about 0.5 K warmer. Though other airborne IR images did not have coincident ASTER observations for comparison, there was an apparent brightness temperature gradient present in most of the images from east-west flight tracks, but not in any from north-south flight tracks.

The apparent brightness temperature gradients could be due to the temperature of the thermal infrared cameras themselves, view angle effects as described in Section 2.2, or a combination of the two. Since the airborne IR image mosaics are created from individual images from three cameras, these differences may stem from the three cameras having different temperatures. Following the east-west flightlines, solar heating (sunlit on the southern side, shaded on the northern side) on the aircraft could contribute camera temperature differences (Pestana et al., 2019). Likewise, prevailing wind direction could preferentially cool

one side of the camera set, versus the other. Though centered at nadir, the three airborne IR cameras together have view angles from 31.5° on the left to 41.0° on the right. At larger off-nadir view angles near the image edges more of the sides of trees will be visible. Along the east-west flightlines across, the north looking cameras are seeing the south-facing and sunlit side of trees and fewer tree shadows, an overall slightly warmer scene than a nadir view would provide. At the same time, south looking cameras are seeing the north-facing side of trees and more snow surface shaded by the trees, an overall slightly colder scene

than a nadir view would provide. A similar effect could be taking place with undulations in the snow surface where at different view angles the cameras see either the warmer south-facing sunlit side or cooler shaded north-facing side of waves or dunes in the snow surface (Figure 3b).

Whatever the source of the apparent brightness temperature gradient in some of the airborne IR imagery, we considered its impact on the results as negligible. The gradient only had a magnitude of 1 K across the image swaths, which is less than the

cameras' accuracy of ±2 K (Table 1). The difference between airborne IR and ASTER brightness temperatures in the nadir-looking center of the swaths is 0 K. The east-west flightlines passed directly over most of the ground observations locations, observing them from very close to nadir where airborne IR and ASTER brightness temperatures agree  (Figure 1). In comparisons between GOES ABI and airborne IR infrared brightness temperatures, the airborne IR images are aggregated to a mean brightness temperature within a GOES ABI pixel footprint across the image swaths (Figure 4). This aggregation may

compensate for the erroneous gradients since it averages together the positively and negatively biased edges of the swaths. Also of note, the easternmost portion of the airborne IR mosaic from 11:19 shows a large cold feature, which, by inspecting the visible airborne imagery, we identified as a small cloud (Figure 5c), not visible in the airborne or ASTER image from 12 minutes prior. This portion of the mosaic was excluded from later analysis.

**5.2 Comparison of airborne IR, ASTER and ground observations**

Snow brightness temperatures observed by the airborne IR and ASTER imagers were biased warm in comparison with the ground-based snow brightness temperature observations at snow pit #2S10. Aircraft flights on two cloud-free days during the study period provided us with 15 overpasses over the ground sites on the western mesa. Two flights were made on 8 February (Figure 6a,b). On the first flight, three overpasses occurred about an hour after sunrise (7:13 UTC-7) between 8:00 and 9:30. The second flight of that day made six overpasses between 10:30 and 13:00. The second flight was coincident with the ASTER

image taken at about 11:08. On 11 February, a single flight made another six overpasses of the ground site between 9:30 and 13:00 (Figure 6c,d).

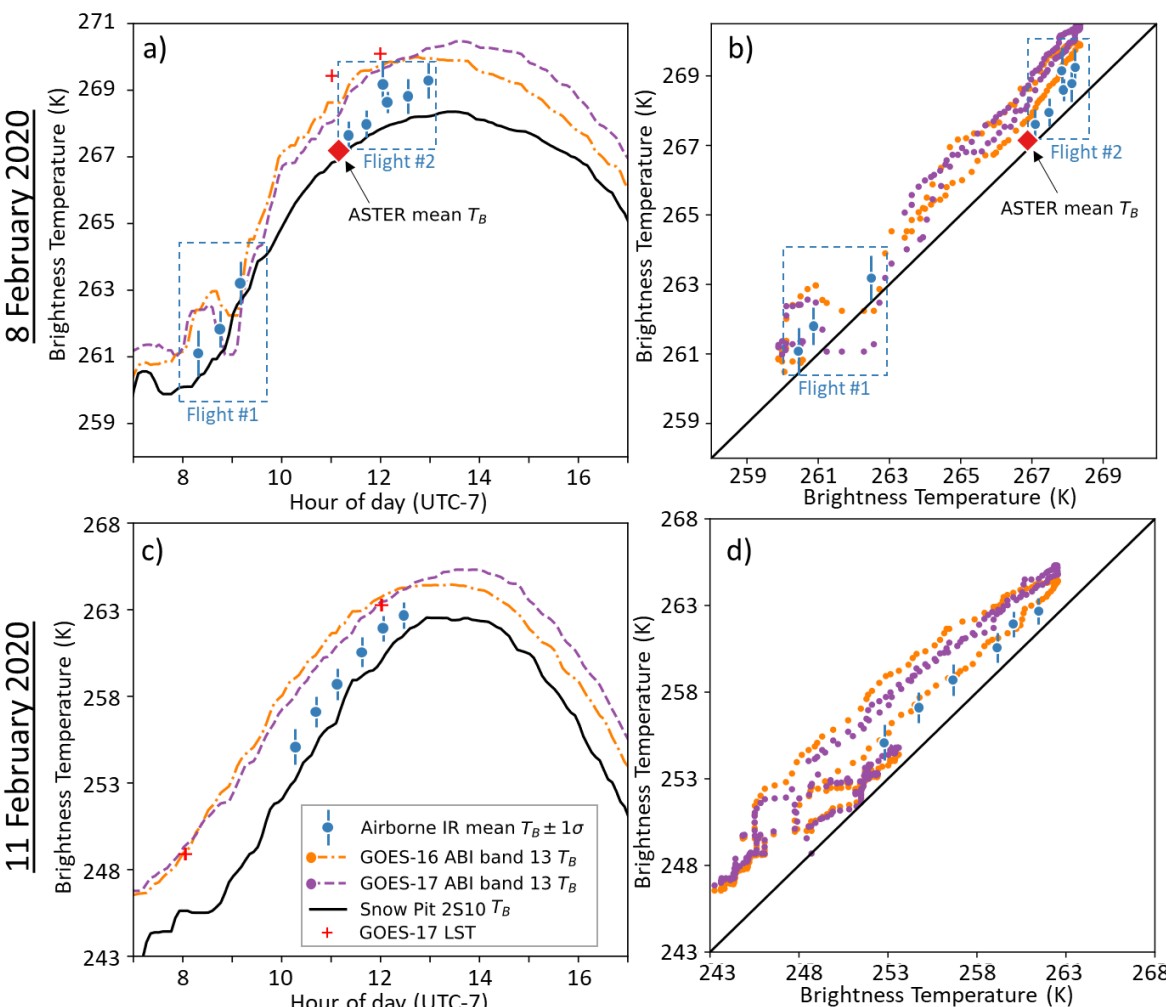

**Figure 6. Timeseries for a) 8 Feb. and c) 11 Feb. of the ground-based snow brightness temperatures from the Apogee radiometer at snow pit #2S10 (black line), along with GOES-16 (dashed orange) and GOES-17 (dashed purple) band 13 brightness temperatures, GOES-17 LST (red +), airborne IR (blue circles), and ASTER (red diamonds) mean brightness temperatures for the 1 km² area around the ground site. Plots of ground-based snow brightness temperature against remote sensing brightness temperatures on b) 8 Feb. and d) 11 Feb.**

On 8 Feb., snow brightness temperatures as seen by the airborne IR and ASTER imagery appeared more uniform than on 11 Feb. around snow pit #2S10, and therefore less sensitive to the size of the area sampled from the imagery to compare with ground-based observations (Table 2). ASTER on 8 February matched most closely to the ground-based brightness temperatures with its single pixel value, though this difference only increased by ~ 0.3 K as the size of the sampled region increased. Snow brightness temperatures as seen in the airborne IR imagery were more uniform across the study area on 8 February (with standard deviations across the 1 km² area of 0.2 to 0.6 K) than on 11 February (with standard deviations of 0.7 to 1.0 K) which had colder air temperatures and higher wind speeds.

**Table 2. Mean difference between ground-based snow brightness temperatures and remotely sensed brightness temperatures (K) from both ASTER and airborne IR imagery, sampled from square areas with sides 5 to 1000 m.**

| Box size of image area sampled (m) | Mean Difference (K) with Ground-based Tb at Snow Pit 2S10 | | |
|---|---|---|---|
| | ASTER, 8 Feb. | Airborne IR, 8 Feb. | Airborne IR, 11 Feb. |
| 1000 | 0.3 | 0.6 | 2.0 |
| 500 | 0.3 | 0.9 | 2.3 |
| 250 | 0.1 | 0.9 | 2.0 |
| 100 | 0.0 | 0.9 | 1.8 |
| 50 | - | 0.9 | 1.7 |
| 25 | - | 0.9 | 1.5 |
| 10 | - | 0.9 | 1.6 |
| 5 | - | 1.0 | 1.4 |

Only four snow pits on 8 February (1N1, 1S2, 6N16, 2S7), and two snow pits on 11 February (1N3, 2S6) were captured in the airborne IR imagery within +/-30 minutes of their snow surface temperature measurements. On the 8th, two of the snow pit surface temperature measurements (1S2, 2S7) were within +/- 1 K of the brightness temperatures in the airborne IR images, two (1N1, 6N16) were ~2 K warmer, and on 11 February, both snow pit temperature observations were 3 – 4 K warmer. Measuring the temperature of the top-most centimeter of snow is not trivial, as the stem thermometers used can heat up if exposed in sunlight, and are in contact with both snow grains and the air space between snow grains. Near the top of the snowpack, the air temperature can be close to that of the above-surface air (Colbeck, 1989), potentially biasing these snow surface temperature readings more towards that of warmer ambient air temperatures.

### 5.3 Comparison of high temporal resolution GOES-R ABI with continuous ground observations

Both GOES-16 and -17 reported surface brightness temperatures warmer than the ground-based snow brightness temperature observations (Figure 6), though this difference varied over the course of each day (Figure 7). Compared to the ground-based observations, the band 14 brightness temperatures had smaller mean and root-mean-squared differences than did the band 13 brightness temperatures. ABI brightness temperatures and ground-based temperature observations show a hysteresis patterns, with GOES ABI brightness temperatures more closely matching the ground-based observations in the nighttime (11 Feb.) and early morning (8 Feb.) than during the day. This pattern is more apparent on 11 Feb., with GOES brightness temperatures warming up in the morning, and cooling down in the evening, faster than the ground-based snow brightness temperatures.

GOES-16 and -17, bands 13 and 14, all observed daily $T_{min}$ and $T_{max}$ within 1 hour or less of those measured on the ground, and the DTR matched within +/- 3 K on both days. On 8 February, GOES-16 and -17, bands 13 and 14, all observed $T_{max}$ within 30 minutes of ground-based $T_{max}$. Both GOES-16 and -17 observed a DTR ~3 K larger in this time period than the DTR measured on the ground. On 11 February, GOES-16 bands 13 and 14 observed the time of $T_{max}$ 30 minutes later than the ground-based $T_{max}$, and $T_{min}$ within 15 minutes. GOES-17 saw $T_{max}$ almost 1 hour later than ground-based $T_{max}$, and a $T_{min}$

within 20 minutes of ground-based $T_{min}$. On this day, the DTR observed by GOES ABI was ~3 K smaller than the DTR from the ground-based snow brightness temperature observations.

### 5.4 Comparison of GOES-R ABI, airborne IR, and ASTER imagery

The band 13 and 14 brightness temperatures from both GOES-16 and -17 were mostly warmer than those from airborne IR and ASTER imagery, and this warm bias was larger for GOES-16, especially for ABI pixels which contained larger forest fractions. The mean differences between GOES-16, compared to GOES-17 and airborne IR observations, across all pixels during the first flight on 8 February decreased over time from a positive to a negative biases (Figure 7a,b). During the second flight on 8 February, the mean differences between GOES-16 and airborne IR generally decreased from about 2 K to 0 K, while the mean differences between GOES-17 and airborne IR increased over time from -1 K to 1 K. On 11 February, the mean differences between GOES-17 and airborne IR were relatively constant throughout the morning of observations, while GOES-16 mean differences decreased similarly to what was seen on 8 February, from 5 K to 2 K (Figure 7c,d). The brightness temperature differences also showed some variation due to the airborne image swaths sampling only part of the ABI pixel footprints on alternating flight tracks (as seen in Figure 7b,d). The mean differences between GOES-16 and the morning airborne IR observations were found to be larger for ABI pixel footprints with higher $f_{veg}$, while the differences with GOES-17 did not correlate with $f_{veg}$ (Figure 8). Similarly, the mean differences between GOES-16, but not GOES-17, and the morning ASTER observation were larger for ABI pixel footprints with larger $f_{veg}$. These results were also robust to uncertainty in the geolocation accuracy of GOES ABI pixel footprints, which is discussed in Section 6.1.

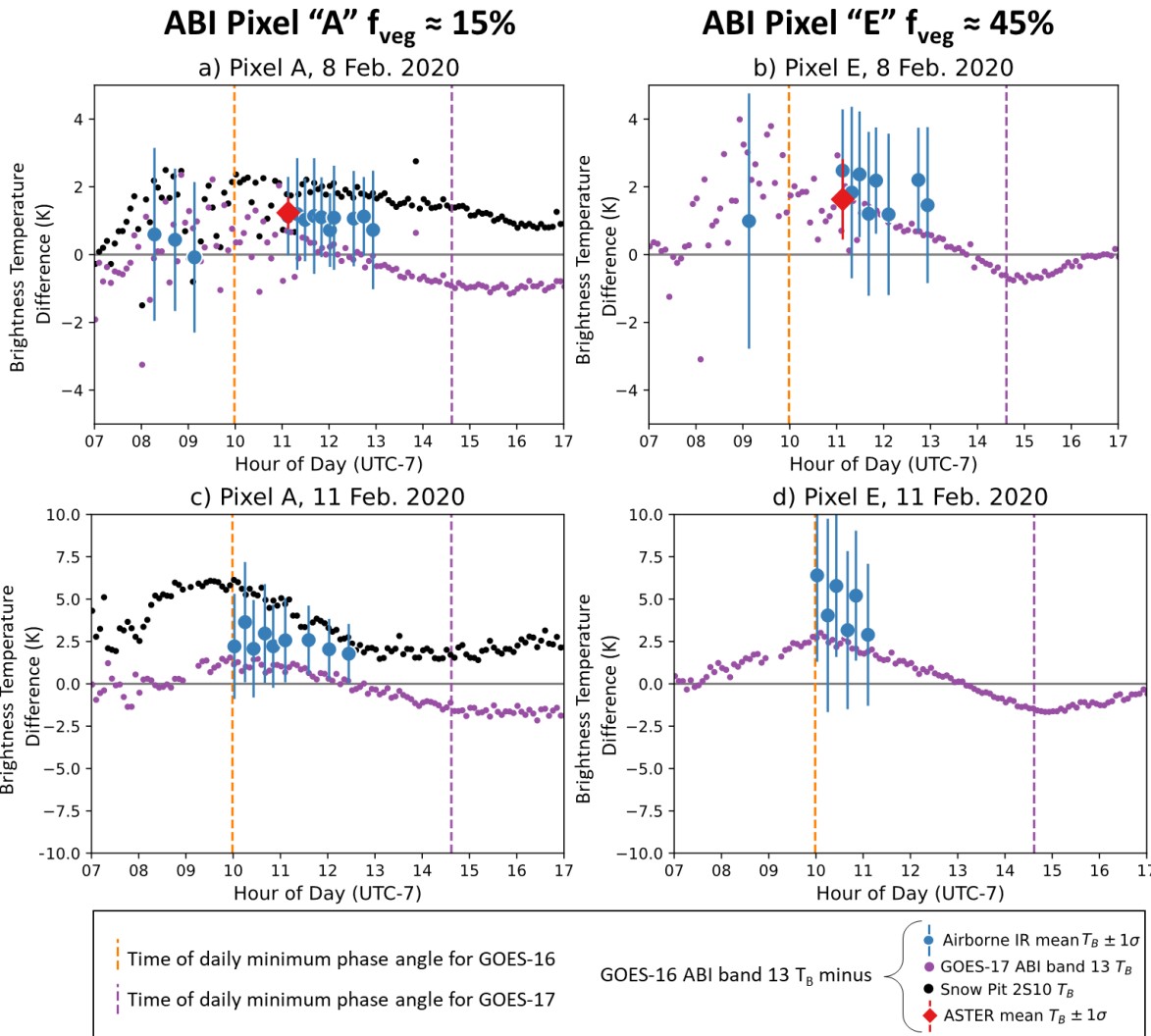

**Figure 7. Difference between GOES-16 ABI band 13 brightness temperature, and brightness temperature observations from GOES-17 ABI band 13, airborne IR imagery, ASTER imagery, and ground-based brightness temperature observations. Plots for (a,b) 8 February 2020 and (c,d) 11 February 2020 for pixels (a,c) A ($f_{veg}$~15%) and (b,d) E ($f_{veg}$~45%). The times that GOES-16 and GOES-17 have their daily minimum phase angle are marked with vertical dashed orange and purple lines, respectively.**

The difference between GOES-16 and -17 brightness temperatures showed a prominent pattern over the course of each day, with GOES-16 reporting warmer brightness temperatures by as much as 3 K in the morning, peaking at about 10:00, and GOES-17 reporting warmer brightness temperatures by nearly 3 K in the afternoons, peaking at about 15:00. The maximum and minimum differences between GOES-16 and -17 were larger for the more forested pixel E than the mostly open snow pixel A. However, there was no correlation found between the magnitude of these differences and the $f_{veg}$ value of each pixel.

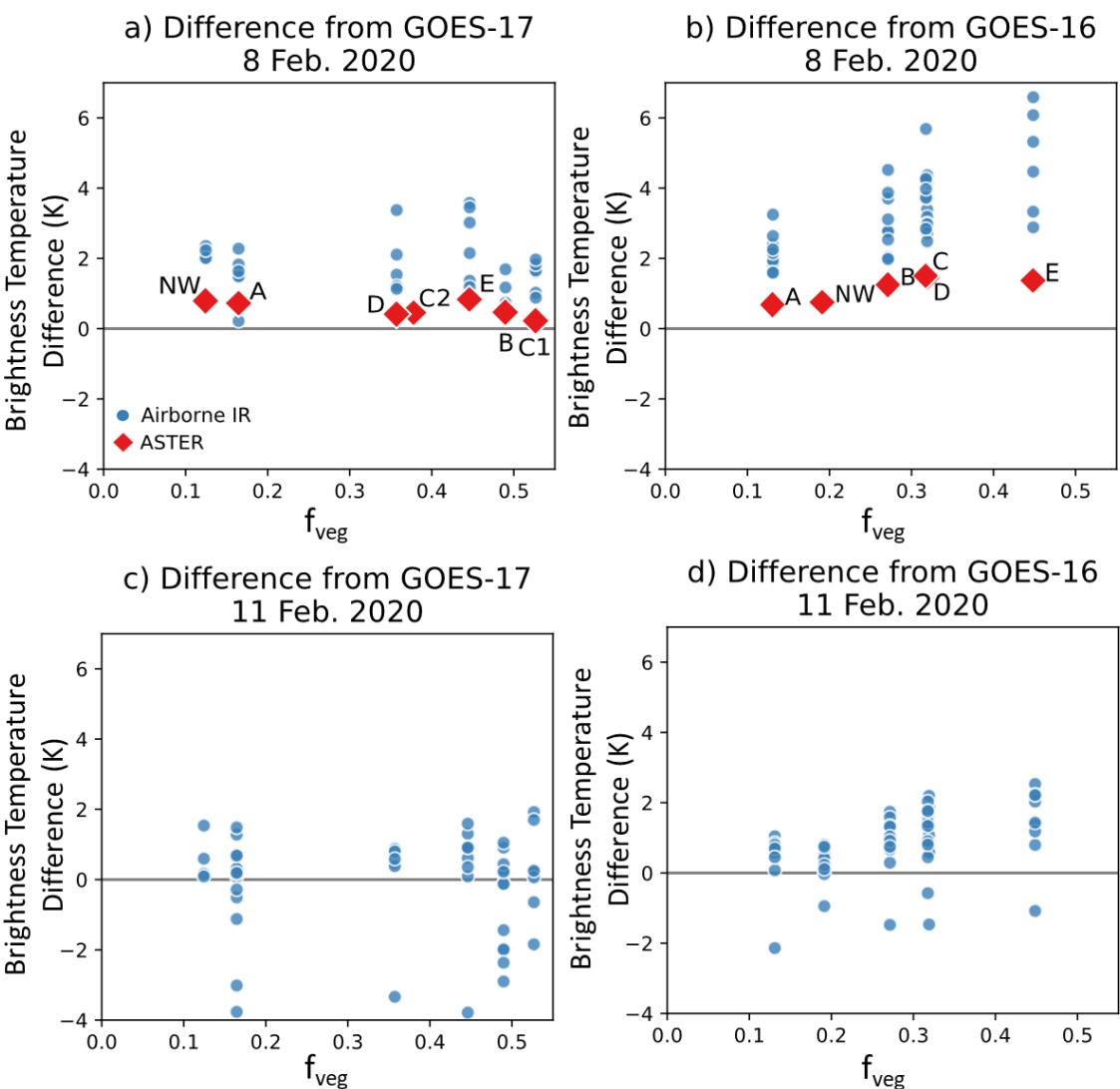

**Figure 8. Mean differences between GOES-16 and -17 ABI brightness temperatures, airborne IR (blue dots) and ASTER (red diamonds), plotted against the fractional vegetated area (f_veg) value of each ABI pixel footprint.**

## 6 Discussion

### 6.1 Intercomparison of remote sensing data

The mean differences between brightness temperatures observed by all remote sensing sources and ground-based observations ranged from about 0 to 5 K, with remote sensing sources (airborne IR, ASTER, GOES ABI) typically reporting warmer brightness temperatures than those measured on the ground. The airborne IR and ASTER images best matched the ground-based snow brightness temperature observations because they could resolve separate snow and forest temperatures, whereas

the coarser 2+ km spatial resolution GOES-R ABI pixels reported a mixture of forest and snow temperatures (Table 3). This was apparent even for the mostly forest-free westernmost portion of Grand Mesa (pixels A) with $f_{veg} \approx 15\%$. Additionally, there were some thin high-altitude clouds on the morning of 8 February, visible in the GOES-R ABI near-infrared "cirrus band" (band 4, 1.37 µm). The timeseries of band 13 and 14 brightness temperatures during this time show rapid changes, and possibly colder temperatures than would be reported if those thin clouds had not been present.

**Table 3. Summary of mean differences (K) between the various brightness temperature (Tb) data sources (aggregated across all ABI pixel footprints where applicable) for two days of coincident observations during the SnowEx 2020 field campaign. Cells are colored by sign and magnitude of the difference, with positive differences in red and negative differences in blue.**

| | Ground Tb | Airborne IR Tb | ASTER Tb | GOES-17 b13 Tb | GOES-17 b14 Tb | GOES-16 b13 Tb | GOES-16 b14 Tb | |
|---|---|---|---|---|---|---|---|---|
| **Ground Tb** | | 0.8 | 0.3 | 1.6 | 1.2 | 1.3 | 1 | |
| **Airborne IR Tb** | 1.8 | | 0.6 | 0.3 | 0 | 1.1 | 0.9 | **8 February 2020** (difference (K) = column − row) |
| **ASTER Tb** | - | - | | 2.2 | 0.6 | 2.6 | 1.2 | |
| **GOES-17 b13 Tb** | 2.9 | 1.7 | - | | -0.3 | 0.1 | 0.3 | |
| **GOES-17 b14 Tb** | 2.7 | 1.6 | - | -0.2 | | -0.2 | 0 | |
| **GOES-16 b13 Tb** | 2.7 | 3.4 | - | 0.3 | 0.1 | | -0.2 | |
| **GOES-16 b14 Tb** | 2.4 | 3.5 | - | 0.5 | 0.3 | 0 | | |

**11 February 2020**
(difference (K) = row − column)

Over the course of each morning, the airborne IR imagery tracked the morning warm-up of the snow surface closely, with a constant warm bias relative to the ground-based observations (Figure 6). GOES-16 and -17 ABI also tracked the ground-based snow brightness temperature observations over the course of the day, though their biases relative to those observations changed over time (Figure 7). Snow brightness temperatures were more uniform across the western mesa on 8 February than on 11 February, as seen in airborne imagery (Table 2). This was also reflected in ABI brightness temperatures more closely matching the ground-based observations on 8 February. Both GOES-16 and -17 captured the timing of daily $T_{min}$ and $T_{max}$ on these two days within ~1 hour, and the diurnal temperature range within +/- 3 K. This uncertainty in the DTR is similar to the range seen in the mean differences over each day and across the mesa between GOES-R ABI and the ground-based observations (1-3 K), ABI and ASTER (2 – 3 K), and ABI and the airborne IR imagery (0 – 5 K).

Our use of a ground-based infrared radiometer, and thermal infrared imagers all within the 8-14 µm window allowed us to directly compare their observed brightness temperatures (Pestana and Lundquist, 2022). However, the radiometer and imagers viewed the study site through different atmospheric path lengths. These different path lengths would subject the

observations to different amounts of atmospheric absorption and emission of infrared radiation by water vapor. This impact is somewhat alleviated at our high-elevation continental study site, where the atmospheric path to a satellite imager is shorter and atmospheric water vapor concentrations are less than those at more coastal or lower elevations. To quantify these effects, the impact of atmospheric path length for the airborne and satellite thermal infrared imagers was simulated with MODTRAN

(Berk et al., 2014) for a mid-latitude winter atmosphere. These simulations showed that GOES-R ABI brightness temperature observations could be as much as 4 K colder than true brightness temperatures due to atmospheric absorption. The GOES-R ABI LST product, which is designed to account for these atmospheric effects, reported surface temperatures very close to GOES-R ABI brightness temperatures (Figure 6). Absorption by water vapor along the atmospheric path between the snow surface and the radiometer mounted < 2 m above the snow surface is negligible; however, for the airborne

IR observations with a path length of ~1 km, the MODTRAN simulation showed that brightness temperatures could be as much as 2 K colder than true brightness temperatures. Due to their different atmospheric path lengths alone, we would expect satellite observations of top-of-atmosphere brightness temperature to be biased colder than airborne observations from ~1 km. Our results, however, show that GOES-R ABI brightness temperatures were biased warmer than airborne observations, suggesting that the magnitude of the atmospheric effect is surpassed by view angle related effects.

The mean differences between ground-based snow brightness temperature observations and ABI band 14 brightness temperatures were smaller than those for band 13 by about 0.2-0.4 K. This, however, doesn't necessarily mean that band 14 was providing a more accurate snow brightness temperature reading. Band 14 covers wavelengths where we can expect some absorption of infrared radiance by atmospheric water vapor, whereas band 13 sits within the "clean IR" window with minimal to no infrared absorption by water vapor (Schmit et al., 2018). With no atmospheric water vapor absorption, the

difference between band 13 and 14 brightness temperatures would be negligible. Any atmospheric water vapor present can result in band 14 brightness temperatures being colder than band 13, as was the case seen here. Though we chose generally cloud-free time periods of observations, any trace amounts of water vapor could be causing band 14 to appear colder, and which coincidentally more closely matched higher-resolution snow brightness temperatures on Grand Mesa.

The results of this intercomparison were robust even if the geolocation accuracy of a GOES ABI pixel footprint was an order

of magnitude larger (± 500 m) than their reported accuracy (±50 m; Tan et al., 2018). To test the sensitivity of our results to geolocation accuracy, the analysis was repeated but with airborne and ASTER imagery clipped to the area of expanded GOES ABI pixel footprints. These expanded footprints included a 500 m buffer around the original footprint perimeter (Figure 4b), making them 1 km wider and taller than the original pixel footprints, representing large location uncertainty and overlap between adjacent pixel fields of view. This also tested the assumption that the radiance detected by GOES ABI for a given

pixel originated only from the land surface contained within the pixel footprint (i.e. that true brightness temperatures, represented by 5 m airborne or 90 m ASTER, should linearly scale to an average brightness temperature detected by GOES ABI). The analysis using these expanded footprints did not significantly change the results of this work. The warm biases seen in GOES ABI brightness temperatures were larger, likely due to the expanded pixel footprints including warmer lower elevation areas off the edge of Grand Mesa (Figure 2).

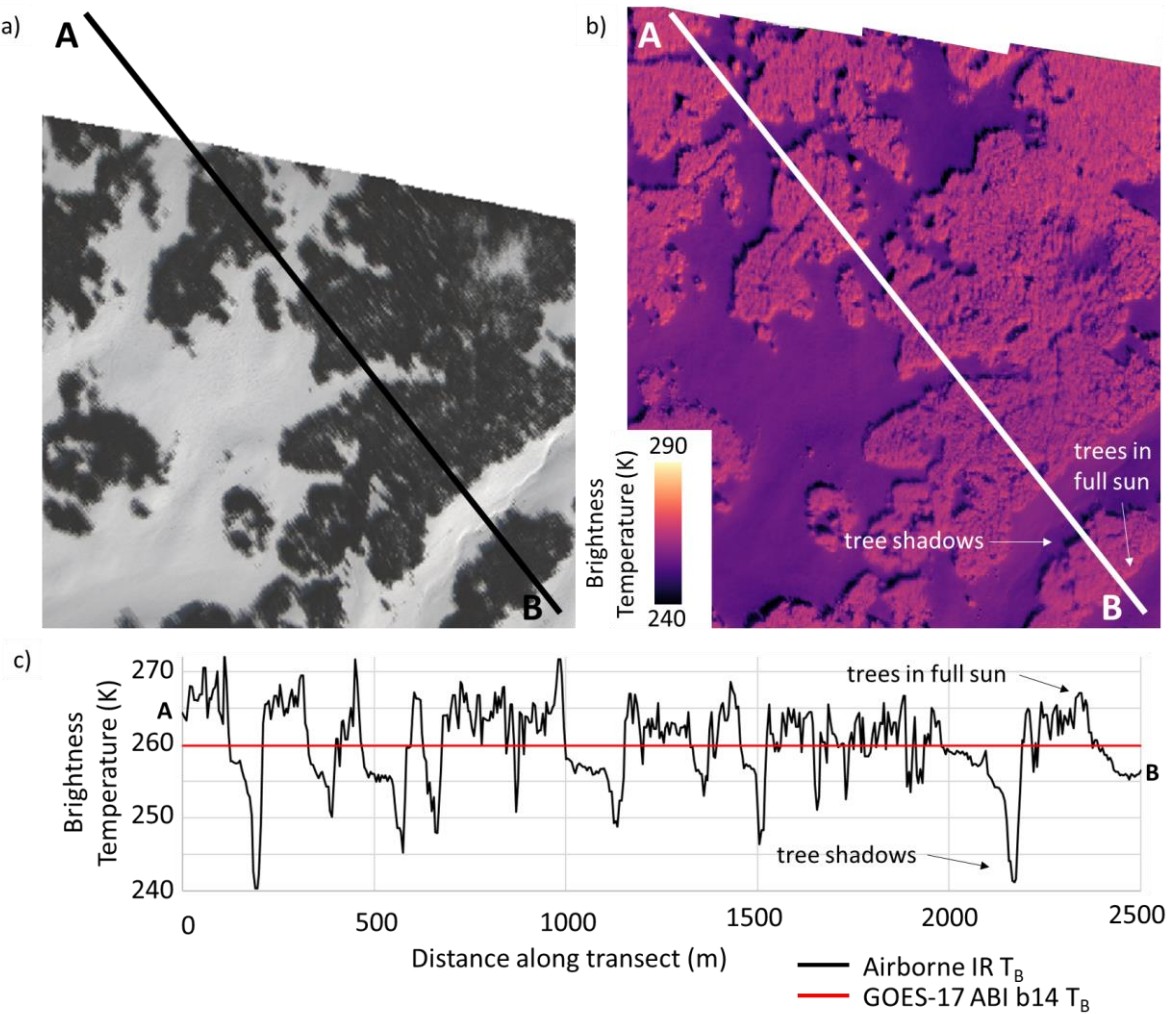

**Figure 9. Approximately nadir airborne a) visible and b) IR images over Grand Mesa, Colorado from 2020-02-11 10:25:51 UTC-7, and c) a temperature profile across a forest stand, showing the presence of very cold tree shadows, and warm southeast forest edges in full sun. The temperature profile is parallel with the view direction of GOES-16, and nearly perpendicular to the view direction of GOES-17.**

## 6.2 Sun-satellite phase angle and thermal infrared shadow-hiding

Even with the flat terrain of Grand Mesa controlling for effects of viewing mountain terrain from off-nadir angles (Pestana and Lundquist, 2022), we observed a morning warm bias between GOES-16 and the coincident nadir-looking ASTER, airborne IR imagery, and GOES-17 (Figure 7). The GOES-16 brightness temperatures were potentially exhibiting a hotspot effect when the angle between the sun and the view angle of GOES-16 (phase angle) reached a daily minimum. The hotspot effect seen in remote sensing imagery of forests is understood to be explained by shadow-hiding in imagery of reflected sunlight in the visible and NIR wavelengths (Deering et al., 1999; Hapke et al., 1996).

When the airborne IR and ASTER images were taken (7:00-13:00), the sun was rising in the southeastern sky, in the same direction from which GOES-16 was viewing Grand Mesa. At about 10:00 on 11 February, when we see the largest warm biases between GOES-16 and all other datasets, the sun had reached 26.9 degrees elevation at an azimuth of 139.1 degrees (26.1 and 139.6 on the 8th). The angle between GOES-16's view and the sun's position (phase angle) was at its minimum of ~ 8 degrees on 11 February at 9:59 (~9 degrees at the same time on 8 February). At this time, the sun was illuminating and warming the southeastern facing sides of trees that GOES-16 is viewing, which in the airborne IR imagery were as much as 5 K warmer than the shaded side of trees (Figure 9). In addition to viewing the sunlit side of trees, snow in tree shadows was considerably colder than snow in the sunlight (by ~10 K) and would also be hidden from the view of GOES-16.

The airborne IR and ASTER images viewed the study area from nadir, and the difference between these two image sources did not vary with $f_{veg}$. GOES-17 surface brightness temperatures had smaller mean differences compared to airborne and ASTER than GOES-16, and these differences did not correlate with $f_{veg}$. GOES-17, viewing Grand Mesa from the southwest, would similarly be viewing the southwest facing sides of trees, though during the morning these would be partially in sun and partially in shade. In the afternoon we see that GOES-17 is warmer than GOES-16, peaking at about 15:00. The minimum phase angle between the sun and GOES 17 is ~8 degrees at 14:37 on 11 February (~9 degrees on 8 Feb.).

Though we see that the warm bias in GOES-16 imagery correlates with $f_{veg}$, the presence of these same warm biases and their patterns over time (e.g. warm biases peaking at the time of minimum phase angle) in the mostly open snow pixels (Figure 7a,c) suggests that other sources of surface roughness may also be contributing to this effect, such as greater than meter-scale dunes, or sub-meter-scale ripples and sastrugi (Kochanski et al., 2019; Warren et al., 1998).

## 6.3 Applications for downscaling GOES-R ABI thermal infrared imagery

Downscaling methods for coarse spatial resolution thermal infrared imagery rely on finer spatial resolution maps of land cover properties and statistical relationships to model and therefore correct for the expected biases in the coarse imagery. Prior methods have used vegetation (Inamdar and French, 2009; Kustas et al., 2003) and terrain maps (Walters, 2013), and biases in GOES-16 ABI imagery have been related to their off-nadir views of complex terrain (Pestana and Lundquist, 2022). Our results demonstrate that for high temporal resolution GOES-R ABI thermal infrared imagery, not only does the fractional forest coverage of each ABI pixel have some control on surface temperature biases, but so does the solar illumination angle, and the phase angle between the satellite and sun. Thus, any downscaling of GOES-R data must explicitly consider time of day and time of year. These solar and satellite view angle controls on surface temperature observation biases were observed over both the forested and open snow regions of Grand Mesa, suggesting that surface roughness features as large as trees but perhaps as small as sastrugi contributed to the hotspot effect seen. This information will be needed to determine if, when, and what magnitude a hotspot or thermal infrared shadow-hiding effect will have on the surface temperature bias of the coarser resolution GOES-R ABI.

# 7 Conclusions

During the NASA SnowEx field campaign in February 2020, we conducted an intercomparison of thermal infrared remote sensors for retrieving surface brightness temperatures of snow and forests. The flat study site at Grand Mesa in western Colorado, USA, allowed us to investigate the impact that forest cover has on thermal infrared remote sensing from GOES-16 and GOES-17 at off-nadir view angles and high temporal resolution. Snow brightness temperatures observed by the airborne IR and ASTER imagers were biased warm in comparison with the ground-based snow brightness temperature observations, and the airborne IR imagery itself was found to have a warm bias compared with ASTER, all with mean differences within < 1 K of each other. GOES-16 and GOES-17 observed daily maximum and minimum brightness temperatures within ~1 hour of those measured in situ, and the diurnal temperature range matched within +/- 3 K. GOES-16 and GOES-17 reported warmer surface brightness temperatures than the ground-based, airborne IR, and ASTER observations. This warm bias was larger for GOES-16 in the mornings when the aircraft and ASTER passed over the study site. The maximum warm biases in GOES-16 and GOES-17 occurred when the sun-satellite phase angle was at its daily minimum, suggesting that a thermal infrared shadow-hiding effect may cause these off-nadir imagers to sense warmer temperatures than nadir-looking imagers. Therefore, land surface roughness features such as trees, and the diurnal changes in phase angle should be considered when interpreting GOES-R ABI observations of land surface or brightness temperatures.

The thermal infrared imagery and ground-based snow temperature observations collected as part of SnowEx 2020 provide a unique dataset for characterizing the high temporal resolution observations from geostationary satellites. It could be used further for testing methods for spatially downscaling coarse GOES-R ABI imagery of snow and forests to finer spatial resolutions with statistical models, sensor fusion methods (Quan et al., 2018; Weng and Fu, 2014), or using spectral mixture models to separate snow and forest temperatures (Lundquist et al., 2018). This work demonstrates that future applications of GOES-R ABI imagery for land surface temperature observations of landscapes like mountain snow and forests must account for the continuously changing phase angle and resulting thermal infrared shadow hiding at small phase angles. Though this work focuses on a single site in a short time period, other geostationary satellites comparable to GOES-R ABI, such as Fengyun-4 and Himawari-8, provide similar views of High Mountain Asia and other mountains in the Eastern hemisphere where these observations are needed. We expect the processes described here to be important for interpreting geostationary thermal infrared observations all around the globe.

# 8 Code availability

The code used in the analysis of these datasets, including python scripts and Jupyter Notebooks to generate plots and figures, are available at https://github.com/spestana/snowex2020/tree/v1.0 (Pestana, 2023)

## 9 Data availability

All data used in this work is publicly accessible. The continuous snow brightness temperature observations (Pestana and Lundquist, 2021), instantaneous snow surface temperatures (Vuyovich et al., 2021; Johnson et al., 2023), and airborne IR imagery (Chickadel et al., 2022) are available through the National Snow and Ice Data Center (NSIDC). The ASTER imagery is accessible through the USGS LPDAAC (https://lpdaac.usgs.gov/), and NOAA Geostationary Operational Environmental Satellites (GOES) 16 & 17 imagery from https://registry.opendata.aws/noaa-goes/ with goespy (Mello and Pestana, 2022).

## 10 Competing interests

The contact author has declared that none of the authors has any competing interests.

## 11 Author contributions

All authors provided conceptualization, methodology design, data collection and curation, and manuscript editing; SP performed the analyses, created visualizations, and prepared the manuscript draft with contributions from JL and CC.

## 12 Acknowledgements

We would like to thank the SnowEx 2020 organizing team, participants, snowmobile guides, National Snow and Ice Data Center staff, and the Naval Postgraduate School pilots for facilitating the data collection, and the SWESARR team for accommodating our work and providing IMU/GPS navigation data. We would also like to thank fellow project team members from SnowEx Hackweek 2021 for their work developing workflows and tools to analyze the SnowEx snow temperature observations: Aji John, Jeremy Johnston, Friedrich Knuth, Jewell Lund, Giulia Mazzotti, Zachary Miller, Wenge Ni-Meister, Dillon Ragar. Thank you to the University of Washington Mountain Hydrology group for feedback and input throughout this project, and to our three anonymous reviewers whose feedback and suggestions were very helpful in improving this paper. This work was funded by NASA FINESST grant 80NSSC20K1610 and NASA grant 80NSSC20K0374.

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
