# Peer review of "Thermal infrared shadow-hiding in GOES-R ABI imagery: snow and forest temperature observations from the SnowEx 2020 Grand Mesa field campaign"

_EGUsphere, 2023_

## Referee Comment (RC1)

Referee comment on "Thermal infrared shadow-hiding in GOES-R ABI imagery: snow and forest temperature observations from the SnowEx 2020 Grand Mesa field campaign" by Pestena et al.

The authors analyzed satellite, airborne, and ground-based observations of thermal infrared brightness temperature over a snow- and forest-covered area to investigate the dependence of measured brightness temperature on forest coverage and observation angle. A warm bias in GOES surface brightness temperatures was found compared to nadir-looking airborne and satellite-based ASTER observations. This bias is particularly pronounced when the warmer sunlit sides of the trees were observed, which the authors refer to as the "thermal infrared shadow-hiding effect."

Overall, the manuscript is well suited for publication in The Cryosphere. However, some comments might be considered before.

General Comments

For their study, the authors use different thermal infrared sensors, some of which measure in different wavelength bands. Further, atmospheric correction to derive the surface temperature is not applied. Potential effects were shortly discussed. However, I would recommend performing additional radiative transfer simulations to quantify these effects. What I definitely miss is a detailed discussion of the effects of emissivity on the measured brightness temperature. Please take a look at the paper by Hori et al (2006). There, the authors show the dependence of emissivity on wavelength, snow type, and viewing angle.

Specific Comments

1. Please make sure that you specify the temperature term in the text. Use "brightness temperature" consistently if that is the quantity you mean. "Surface temperature" should be corrected for atmospheric contribution and emissivity. Sometimes I wasn't sure what quantity the text was referring to.
2. P1,L15: "observations collected as part of the NASA SnowEx field campaign in February 2020 provided a rich dataset for comparison. Observations over the course of two cloud-free days spanned the entire study site.": I would not call a study consisting of two days of measurements a "rich" data set. Rather, it is a case study.
3. P2,L36: "as GOES-R ABI": Introduce the abbreviation ABI here.
4. Introduction: The introduction is quite long and should end with the outline of the manuscript. Here, two subsections follow the outline. Think about including a reduced content of subsections 1.1 and 1.2 in the main body of the introduction. Figure 1 would fit in section 2.
5. P4,L99: "At much smaller spatial scales, similar effects can occur with off-nadir daytime TIR observations of scenes containing forests. Solar illumination, especially at low sun angles, will warm up one side of individual trees or clusters of trees more than the other shaded side. These trees will also cast shadows onto the underlying snow surface, and the snow surface temperature in these shadows can be much colder than snow in sunlight (Figure 2)." This already anticipates some of the results of the study. If this is already known, then point out what is still missing from these previous publications and what you can contribute to close some gaps. Think about to use figure 2 for illustration of your results in a later section.
6. P6,L131: "The high emissivities of both snow and conifer trees provide us with a scene where surface brightness temperatures are close to true surface temperatures (Kim et al., 2018; Warren, 2019)." Please elaborate the dependence of the measured brightness temperature on the emissivity (see general comments).
7. Figure 6: Where does the dark stripe in Fig. 6a come from? Is this from overlapping both sensor measurements? A map showing the phase angles would be nice to see here.

8. P8,L165: "Images from the Advanced Baseline Imager (ABI) onboard GOES-16 and GOES-17 were retrieved …" Mention explicitly that the measured brightness temperature at top of atmosphere is used.
9. P11,L275: "The mean and root mean squared difference between GOES ABI brightness temperatures and the ground-based snow and air temperatures were also computed." Is it really the air temperature you want to look at or is it more the surface skin temperature?
10. P14,L316: "This may explain some of the temperature gradient seen in the east-west flight images." The explanation is quite vague. Figure would fit here for illustration.
11. P14,L320: "Snow surface temperatures observed by the airborne IR and ASTER imagers were biased warm in comparison with the ground-based snow surface temperature observations at snow pit #2S10." Again, is it the brightness temperature or the surface skin temperature what is meant here?
12. Table 3: "Airborne IR Ts": Why is this not a brightness temperature? Did you consider the atmospheric contribution between aircraft and surface?
13. P20,L415: "The thermal infrared brightness temperatures observed by ABI are likely to be colder than the actual surface brightness temperature due to atmospheric absorption of infrared radiation." This could be elaborated further. I encourage the authors to use radiative transfer simulations to investigate the sensitivity more deeply and to make more quantitative statements.
14. P20,L422: "Absorption by water vapor along the atmospheric path between the snow surface and the radiometer mounted < 2 m above the snow surface is negligible; however, for the airborne IR observations with a path length of ~1 km…". Flight altitude should be given earlier.

Technical Comments

1. Y-axis: Specify the "temperature". Is it the brightness temperature or the surface temperature what is displayed here?
2. Table 1: "Ground-based observations" not fully legible.
3. Figure 5: Enlarge the color bar. "Temperature" → "Brightness temperature"
4. Table 2: Give the unit of the differences.
5. P17,L378,L379: "fveg" → "$f_{veg}$"
6. Figure 8 and figure 9: Give temperature differences in units of Kelvin.

References:

Hori, Masahiro & Aoki, Teruo & Tanikawa, Tomonori & Motoyoshi, Hiroki & Hachikubo, Akihiro & Sugiura, Konosuke & Yasunari, Teppei & Eide, Hans & Storvold, Rune & Nakajima, Yukinori & Fumihiro, Takahashi. (2006). In-situ measured spectral directional emissivity of snow and ice in the 8–14 μm atmospheric window. Remote Sensing of Environment. 486-502. 10.1016/j.rse.2005.11.001.

---

## Author Response (AR1)

Dear reviewers,

Thank you again for your very helpful comments and suggestions to improve this paper. All of the original reviewer comments are provided below. Our initial responses are in the first box beneath each comment, and secondary responses after changes to the paper have been made are in the second box noting new Figure or Section numbers, page and line numbers where edits have been made.

Sincerely,

Steven Pestana

**Reviewer #1:**

Referee comment on "Thermal infrared shadow-hiding in GOES-R ABI imagery: snow and forest temperature observations from the SnowEx 2020 Grand Mesa field campaign" by Pestena et al.

The authors analyzed satellite, airborne, and ground-based observations of thermal infrared brightness temperature over a snow- and forest-covered area to investigate the dependence of measured brightness temperature on forest coverage and observation angle. A warm bias in GOES surface brightness temperatures was found compared to nadir-looking airborne and satellite-based ASTER observations. This bias is particularly pronounced when the warmer sunlit sides of the trees were observed, which the authors refer to as the "thermal infrared shadow-hiding effect."

Overall, the manuscript is well suited for publication in The Cryosphere. However, some comments might be considered before.

Dear Reviewer #1,

Thank you for your very thoughtful and constructive review. By addressing your concerns and comments we can greatly improve this paper.

We plan on addressing your primary concerns by 1) clarifying in our discussion section that we did simulate atmospheric absorption with MODTRAN to determine what effect this had on our results, and 2) adding more discussion about surface emissivities and the impact this had on our results. We will also make sure to clarify throughout the paper (including figures and tables) whether we are discussing "brightness" or "surface" temperature, and make sure to define these early on for the reader. Similarly, we will report brightness temperatures in units of Kelvin only (including figures and tables). Finally, we will address each of the individual comments as described below.

Thank you again for taking the time to read and review our paper, and I hope with these proposed changes we can submit a revised manuscript soon!

Sincerely,
Steven Pestana

We have edited the paper to address your primary concerns as follows:
1) We clarified our use of MODTRAN in Section 6.1 to address uncertainties related to atmospheric effects on our thermal infrared remote sensing.

> 2) Emissivity, and especially the impact that variations in angular emissivity of snow could have on our results is now discussed in Section 2.2.
>
> We have clarified our use of the term "brightness temperature" throughout the paper, and report all brightness temperatures in units of Kelvin.

General Comments

For their study, the authors use different thermal infrared sensors, some of which measure in different wavelength bands. Further, atmospheric correction to derive the surface temperature is not applied. Potential effects were shortly discussed. However, I would recommend performing additional radiative transfer simulations to quantify these effects.

> Though we included a brief discussion of the different atmospheric path lengths each sensor viewed the study site though, we agree that including calculations of the actual brightness temperature differences due to atmospheric absorption would help. We plan on expanding the discussion section where this was briefly mentioned to include this additional information (section 5.1).
>
> We have clarified in Section 6.1 our use of MODTRAN to quantify the atmospheric effects on our results.

What I definitely miss is a detailed discussion of the effects of emissivity on the measured brightness temperature. Please take a look at the paper by Hori et al (2006). There, the authors show the dependence of emissivity on wavelength, snow type, and viewing angle.

> Thank you for pointing this out. We only just mentioned surface emissivities in section 2.1 but it does require more attention. We plan on adding further discussion about differences in observations due to the angular dependence of emissivity. This will be added to section 1.2 (see other comments later about moving this to it's own background section) and again in the discussion of our results (section 5.1).
>
> We have added to Section 2.2 a discussion of the anticipated influence of angular emissivity on our intercomparison of different brightness temperature observations.

Specific Comments

1. Please make sure that you specify the temperature term in the text. Use "brightness temperature" consistently if that is the quantity you mean. "Surface temperature" should be corrected for atmospheric contribution and emissivity. Sometimes I wasn't sure what quantity the text was referring to.

   > We will clarify our use of "brightness temperature" or "surface temperature" throughout the paper. We will be careful to fix the terminology (and state within the paper how we are using this terminology) so that brightness temperature refers only to non-atmospherically-corrected observations, and surface temperature refers to atmospherically-corrected observations or kinetic temperatures (such as those measured by in situ contact thermometers).
   >
   > "Brightness temperature" is used throughout the paper now where appropriate (our remote sensing and ground-based radiometer observations).

2. P1,L15: "observations collected as part of the NASA SnowEx field campaign in February 2020 provided a rich dataset for comparison. Observations over the course of two cloud-free days spanned the entire study site.": I would not call a study consisting of two days of measurements a "rich" data set. Rather, it is a case study.

> We will re-word this sentence for clarity:
> "As part of the NASA SnowEx field campaign in February 2020, coincident surface brightness temperature observations from ground-based and airborne IR sensors were collected to compare with those from the geostationary satellites."
>
> This has been reworded (P1, L14)

3. P2,L36: "as GOES-R ABI": Introduce the abbreviation ABI here.

> We'll define Advanced Baseline Imager (ABI) has here as it's the first instance that we use the abbreviation.
>
> This has been fixed (P2, L37)

4. Introduction: The introduction is quite long and should end with the outline of the manuscript. Here, two subsections follow the outline. Think about including a reduced content of subsections 1.1 and 1.2 in the main body of the introduction. Figure 1 would fit in section 2.

> Based on this comment and other reviewer comments, we plan on moving sections 1.1 and 1.2 to their own Background section 2 (now 2.1 and 2.2), and can add a summary of those background sections to the main body of the introduction.
>
> Section 2 was added starting on P3.

5. P4,L99: "At much smaller spatial scales, similar effects can occur with off-nadir daytime TIR observations of scenes containing forests. Solar illumination, especially at low sun angles, will warm up one side of individual trees or clusters of trees more than the other shaded side. These trees will also cast shadows onto the underlying snow surface, and the snow surface temperature in these shadows can be much colder than snow in sunlight (Figure 2)." This already anticipates some of the results of the study. If this is already known, then point out what is still missing from these previous publications and what you can contribute to close some gaps. Think about to use figure 2 for illustration of your results in a later section.

> This section will be edited to clarify what is known about off-nadir thermal infrared satellite observations, and what our work is specifically addressing. Specifically, there is prior work cited that describes the warm temperature bias seen in off-nadir thermal infrared imagery over forests. We will clarify how our work differs from and builds upon this by investigating the effect in high temporal resolution geostationary satellite imagery.
>
> This has been edited and is now part of Section 2.1 (P3).

6. P6,L131: "The high emissivities of both snow and conifer trees provide us with a scene where surface brightness temperatures are close to true surface temperatures (Kim et al., 2018; Warren, 2019)." Please elaborate the dependence of the measured brightness temperature on the emissivity (see general comments).

> We plan on adding further discussion about differences in observations due to the angular dependence of emissivity. This will be added to section 1.2 (see other comments later about

moving this to it's own background section) and again in the discussion of our results (section 5.1).

This has been edited to elaborate on emissivity and is now part of Section 2.2 (starting on P5, L124)

7. Figure 6: Where does the dark stripe in Fig. 6a come from? Is this from overlapping both sensor measurements? A map showing the phase angles would be nice to see here.

This is actually visible airborne imager draped over visible satellite imagery from ASTER. We will update the figure to outline the extent of the airborne imagery (visible image swaths in 6a, infrared images in 6b), include a legend, and clarify this in the figure caption.

This figure has been edited (now Figure 5, P15) with borders around the airborne image swaths.

8. P8,L165: "Images from the Advanced Baseline Imager (ABI) onboard GOES-16 and GOES-17 were retrieved …" Mention explicitly that the measured brightness temperature at top of atmosphere is used.

We'll make sure to mention this here, and in the "Study site and observations" section where our remote sensing data are introduced. We will also make note of this in expanded discussions about the influence of atmospheric absorption on our results (see our responses to prior comments about this).

This has been added in Section 3.3.1, 3.3.3, and in the discussion Section 6.1 (P23, L487)

9. P11,L275: "The mean and root mean squared difference between GOES ABI brightness temperatures and the ground-based snow and air temperatures were also computed." Is it really the air temperature you want to look at or is it more the surface skin temperature?

Thank you for catching this. This was included by mistake (a prior draft of the paper included different analyses with in situ air temperature observations). We will delete this mention of air temperature.

This has been removed (P12, L305)

10. P14,L316: "This may explain some of the temperature gradient seen in the east-west flight images." The explanation is quite vague. Figure would fit here for illustration.

We will re-word this to clarify and better explain the patterns we saw in the airborne imagery. We may also add to Figure 6 to help illustrate our description.

We have expanded the discussion of these patterns in Section 5.1.

11. P14,L320: "Snow surface temperatures observed by the airborne IR and ASTER imagers were biased warm in comparison with the ground-based snow surface temperature observations at snow pit #2S10." Again, is it the brightness temperature or the surface skin temperature what is meant here?

As noted above, we'll make sure to clarify our use of "brightness temperature" or "surface temperature" throughout the paper.

This has been fixed throughout the paper.

12. Table 3: "Airborne IR Ts": Why is this not a brightness temperature? Did you consider the atmospheric contribution between aircraft and surface?

> These will be re-labeled since they are brightness temperatures, not corrected for atmospheric absorption. This will also be corrected in tables 1 and 2 where appropriate.
>
> Labels like these have been fixed to represent brightness temperatures as Tb.

13. P20,L415: "The thermal infrared brightness temperatures observed by ABI are likely to be colder than the actual surface brightness temperature due to atmospheric absorption of infrared radiation." This could be elaborated further. I encourage the authors to use radiative transfer simulations to investigate the sensitivity more deeply and to make more quantitative statements.

> This paragraph does mention the expected differences between non-atmospherically corrected brightness temperature and corrected surface temperatures based on calculations from MODTRAN (Berk et al., 2014), but we failed to mention that we used MODTRAN to calculate these. We will clarify in this section that these are from simulations of infrared absorption through a mid-latitude winter atmosphere.
>
> This has been further clarified in Sections 2.2 and 6.1.

14. P20,L422: "Absorption by water vapor along the atmospheric path between the snow surface and the radiometer mounted < 2 m above the snow surface is negligible; however, for the airborne IR observations with a path length of ~1 km...". Flight altitude should be given earlier.

> Flight altitude was mentioned in the "Study site and observations" section were we describe the airborne imagery collection, but we will be sure to include it in the expanded discussion around atmospheric absorption that we plan to add.
>
> Flight altitude is mentioned again in Section 6.1 (P23, L485)

Technical Comments

1. Y-axis: Specify the "temperature". Is it the brightness temperature or the surface temperature what is displayed here?

> As we clarify "brightness temperature" versus "surface" temperature throughout the paper, we will also correct the axes labels in figures to align with the more precise terminology.
>
> This has been fixed in figures throughout the paper.

2. Table 1: "Ground-based observations" not fully legible.

> We can re-format this table to improve legibility or use abbreviations. If abbreviations are used, we will define them in the table caption.
>
> This table has been edited for legibility, and now includes accuracy metrics in units of K in for easier comparison across data sources.

3. Figure 5: Enlarge the color bar. "Temperature" ⭢ "Brightness temperature"

> We'll enlarge the color bar and correct the temperature label to be consistent with other figures.
>
> The color bar in this figure, now Figure 4, has been edited.

4. Table 2: Give the unit of the differences.

> Units will be added, and based on other comments we will report brightness temperatures (and temperature differences) in Kelvin.

> Units have been added to Table 2.

5. P17,L378,L379: "fveg" ▯ "fveg"

> We will fix the formatting of this subscript throughout the paper (including in figures) to be consistent.

> This has been fixed throughout the paper.

6. Figure 8 and figure 9: Give temperature differences in units of Kelvin.

> Based on this and other comments we will report brightness temperatures (and temperature differences) in Kelvin.

> Brightness temperatures and differences are now all reported in units of Kelvin.

References:

Hori, Masahiro & Aoki, Teruo & Tanikawa, Tomonori & Motoyoshi, Hiroki & Hachikubo, Akihiro & Sugiura, Konosuke & Yasunari, Teppei & Eide, Hans & Storvold, Rune & Nakajima, Yukinori & Fumihiro, Takahashi. (2006). In-situ measured spectral directional emissivity of snow and ice in the 8–14 μm atmospheric window. Remote Sensing of Environment. 486-502. 10.1016/j.rse.2005.11.001.

**Reviewer #2:**

Comments on "Thermal infrared shadow-hiding in GOES-R ABI imagery: snow and forest temperature observations from the SnowEx 2020 Grand Mesa field campaign"

This article compares thermal infrared measurements from satellite, airborne, and ground-based instruments collected in February 2020 during the NASA SnowEx campaign. The authors investigate the effects of forest cover and observation angle on satellite base measurements. The study reveals biases in snow surface temperatures observed by airborne IR and ASTER imagers, with GOES-16 and GOES-17 showing warmer surface brightness temperatures compared to ground-based observations. The warm bias is more pronounced in the mornings for GOES-16. The findings suggest "a thermal infrared shadow-hiding" effect impacting off-nadir imagers.

> Dear Reviewer #2,
>
> Thank you for all of your comments and recommendations. We will address the main concerns you noted with brightness temperature terminology and consistent use of temperature units (Kelvin rather than Celsius for brightness temperatures). Your recommendations for improvements and corrections in the specific comments are very helpful. We plan on addressing each of these as described below and using your recommendations which will improve the quality of the paper.
>
> Thanks again for taking the time to read and review our paper. We hope that our plan to address your comments is acceptable and look forward to submitting a revised manuscript!

Sincerely,
Steven Pestana

The main comments here have been addressed as follows:
1 ) Brightness temperature terminology has been fixed throughout the paper
2) We now report all brightness temperatures and their differences in units of Kelvin

**General comments**

The article is well written and well suited for the Cryosphere. However, I would recommend authors to carefully check the terminology used when discussing temperature measurements, be consistent with the terminology used and possibly also define what they mean with different terms. It seems, for example, that brightness temperature and surface temperature are used interchangeably in some parts of the article which may lead to some confusion. Additionally, be consistent with temperature measurement units.

Thank you for noting this (as did other reviewers). We will clarify throughout the paper (including figures, tables, and captions) whether we are talking about "brightness" or "surface" temperatures, and define in the text what we mean by each (e.g. non-atmospherically corrected, versus atmospherically-corrected or surface kinetic temperature measured in situ). We will also make sure that our use of units for temperature are consistent throughout the paper, and use Kelvin for brightness temperature (including figures, tables, and captions).

This has been addressed as mentioned above.

**Specific comments**

The introduction is quite long. Consider moving sections 1.1 and 1.2 to section 2.

Based on this and other reviewer comments, we plan on moving sections 1.1 and 1.2 to their own Background section 2 (now 2.1 and 2.2).

We have added a new Background Section 2.2.

L165 Advanced Baseline Imager (ABI) - usually abbreviations are explained when first mentioned.

We will define Advanced Baseline Imager (ABI) here as it's the first instance that we use the abbreviation.

This has been edited (P2, L38)

L280 Consider rewording as it may seem that the difference between bands 14 and 13 was calculated, not difference between GOES-16 and GOES-17.

This sentence will be re-worded for clarity. Thank you for catching this.

This sentence has been re-worded as follows (P12, L309): "The differences between GOES-16 and -17 ABI brightness temperatures for 8 February (7:00-18:00) and 11 February (21:00 10 Feb. -18:00 11 Feb.) were computed for each pair of corresponding pixels (NW, A, B, C/C1, C/C2, D, and E) across the mesa."

Figure 7 Consider adding borders etc to figures to clearly identify which parts are from the airborne instrument and which are from the ASTER.

| I think this comment might actually be in reference to Figure 6 (which has airborne visible and infrared imagery overlayed on ASTER satellite imagery). We will add borders and a legend to better show the airborne imagery and describe this in the figure caption.

However, if this is in fact in reference to Figure 7 (timeseries plot of brightness temperature observations), we can add a text label and arrow to point out the airborne observations like we already have for the single ASTER point plotted in 7a and 7b. |
| What is now Figure 5 has been edited to include borders around the airborne IR image swaths. The airborne observations in what is now Figure 6a,b are outlined in boxes to distinguish the two flights on 8 February. |

Table 2 Ground-based snow surface temperature in this table mean radiometer measurements not the snow pit surface temperature measurements discussed in the next section?

| That is correct, what we have labeled as "ground-based snow surface temperature" is actually brightness temperature from the in situ radiometer and not the stem thermometer measurements from other snow pit sites. In addition to clarifying that this is a "brightness temperature" we can re-word the section that follows to better clarify the difference between the two in situ observation types (continuous brightness temperature observations by a tripod mounted radiometer versus instantaneous surface temperature observations with stem thermometers). |
| We have reworded this section to clarify where the radiometer data is used versus the stem thermometers (P18, L398) |

Table 3 This table seems a bit misplaced, it might fit better in the discussion section (where it is mentioned the first time) or be removed altogether. Also, Ts (surface temperature?) and Tb (brightness temperature?) are used in the table without explanation.

| I think moving this table to the discussion section makes sense. And while we are correcting the brightness/surface temperature terminology throughout the paper we'll update this table to be consistent with the text, and define any symbols used in the caption of the table. |
| This table has been moved to Section 6.1 (P22) |

L370 Which band?

| We'll update this sentence to specify the results are for both ABI bands 13 and 14. |
| (P12 ,L311) "This comparison was performed with both ABI bands 14 and 13." |

L378-380 fveg is written differently in different places.

| We will fix the formatting of this subscript throughout the paper (including in figures) to be consistent. |
| This subscript formatting has been fixed throughout the paper and in figures. |

Figure 8 "Difference between GOES-16 ABI band 13 brightness temperature, and surface temperature observations from GOES17 ABI band 13" Do the brightness temperature and surface temperature mean the same thing here?

This is only referring to brightness temperatures and will be corrected. We will be sure to edit this throughout the paper, including here in Figure 8, to use consistent terminology whether we are talking about brightness (non-atmospherically-corrected) or surface (atmospherically-corrected) temperature.

Now Figure 7, this has been fixed to be labeled as brightness temperatures.

L387-389 Consider having times also in UTC-7 to match figure 8.

Thanks for pointing this out. We will change the times (in text and figures) throughout the paper to be in local (UTC-7) time to help the reader have an intuitive understanding of morning versus afternoon observations.

All times are now reported in UTC-7 to aid in interpreting data in the context of morning/afternoon.

L414-427Authors mention that a limited number of ABI LST measurements is available but showing some of the comparisons calculated for this product might still be interesting. Also, additional discussion about atmospheric effects might benefit the article.

We can add in the few LST observations available for another point of comparison, either just in the text of the results section or also in figures like Figure 7. Adding LST to Figure 7 would actually help illustrate some of the points we should make in an expanded discussion about atmospheric effects. Based on your comments and other reviewers' comments, we plan on adding more discussion about this.

Figure 6 now includes some of these LST observations.

**Reviewer #3:**

During the NASA SnowEx field campaign in February 2020, researchers compared thermal infrared remote sensors for measuring surface brightness temperatures of snow and forests at Grand Mesa in Colorado, USA. The study focused on GOES-16 and GOES-17 satellites at off-nadir view angles. Airborne IR and ASTER imagers showed warm biases compared to ground-based observations, with GOES satellites reporting warmer surface temperatures. The warm bias in GOES-16 was more pronounced in the mornings, possibly due to a thermal infrared shadow-hiding effect during the sun-satellite phase angle's daily minimum.

Dear Reviewer #3,

Thank you for your comments and ideas to improve our paper. To address your main points about clarifying sources of uncertainty, we will be expanding on these first in a separate background section, and then later in the discussion section. We had done an alternate analysis challenging the geolocation and linear scaling assumptions (but not included it in this paper) and can include a description of that in a revised paper to help support our overall results. We plan on addressing your other comments as described below.

Thank you for taking the time to read and review our paper, and for providing great suggestions to improve the work. We hope that we've outlined a clear plan to address your comments and concerns and look forward to submitting a revised manuscript!

Sincerely,
Steven Pestana

To address your main comments:
1) We have moved sections into a new Background Section 2 and used this to elaborate on some of the uncertainties you mentioned
2) We have included a discussion on how we tested some of the uncertainties around assumptions we make about scaling and geolocation. This is described in Sections 4.4, 5.4, and 6.1.

While the paper is well written, there is room for further elaboration on the identification and quantification of key sources of uncertainty. Specifically, the examination of four sources demands more comprehensive exploration. Firstly, emissivity, influenced by wavelength, target characteristics, and viewing angles, merits a more detailed investigation (also using RT modeling).

Based on your comments and other reviewer comments, we plan on adding further discussion around the influence that emissivity has on off-nadir observations. We plan on introducing these ideas first in what is currently section 1.2 (which will be moved to a separate background section 2.2) where we only briefly mentioned emissivity. This will be elaborated on in the discussion section to help put our results in context.

Emissivity is now further discussed in Section 2.2.

Secondly, the precision of image geolocation accuracy requires a better characterization.

We can address this by quantifying the (relative) geolocation accuracy of Airborne IR imagery. For the ASTER and GOES ABI products, we do rely on prior literature that investigated those sensors' geolocation accuracies. However we had also done an analysis (not described in the current paper) where we assumed that we didn't know the GOES ABI pixel locations as precisely and that we could not necessarily linearly scale ASTER or Airborne IR imagery at higher spatial resolutions up to GOES spatial resolution. This analysis showed that results did not significantly change if we had 500 m of uncertainty in the precise GOES ABI pixel ground footprint. We can include information about this analysis in the paper to address this point.

These uncertainties and how we tested for them are addressed in Sections 4.4, 5.4, and 6.1.

Thirdly, the impact of atmospheric interactions, not compensated for the various sources of thermal acquisition, can affect satellite images and airborne acquisitions disparately.

> Since we only briefly touched on this, other reviewers also recommended further discussion of the different atmospheric path lengths of the different thermal infrared imagers, and how this may impact our results. We plan on addressing this by clarifying that we did a MODTRAN simulation of thermal infrared remote sensing through a mid-latitude winter atmosphere (section 5.1). We can also introduce this earlier in the paper (such as in the background section).

> We have elaborated on this in Section 6.1.

Lastly, the assumption of a linear combination for upscaling the 5m and 90m images to the resolution of GOES-16 and 17, considering slightly different wavelengths, is implicit. However, the linearity of combining snow and trees, at least for my interpretation, may be questionable. These aspects necessitate further development to enhance the paper's appeal for publication in TC.

> This is a good point, all of our scaling for comparing different resolution imagery did have this implicit assumption. We can address this by stating our assumption explicitly (such as in our methods sections where we describe the scaling), but making sure to discuss how if this assumption is wrong how it might affect our results. As mentioned under the prior comment about geolocation accuracy, we can include information from another analysis (currently not included in this paper) that we had performed. In this alternate analysis, we scaled ASTER and Airborne IR imagery from a region larger than an individual GOES ABI pixel footprint (adding an additional 500 m buffer around the perimeter of the pixel footprint) to test whether imprecise GOES ABI pixel footprints or non-linear scaling might have an effect on our results. These did not change the results enough to change our conclusions. We can add a description of this analysis to the paper to make sure these points are covered sufficiently.

> As mentioned above, this is addressed in Sections 4.4, 5.4, and 6.1.

Detailed comments:

- **Introduction:** Consider incorporating a dedicated "Background" section to shape the introduction more effectively. This section should provide a comprehensive overview of what readers can expect from the intercomparison. Enhance the explanation of potential sources of uncertainty as previously mentioned to provide a clearer context for readers.

  > We will move what are now sections 1.1 and 1.2 to their own Background section (which will include more discussion about the sources of uncertainty you identified).

  > See Section 2.

- **Figure 2:** It would be beneficial for Figure 2 to simulate the actual length of a GOES pixel, offering a more realistic representation. Additionally, include a color bar for temperature to enhance the interpretability of the figure.

  > Thanks for this great idea! We can modify figure 2 to show a much longer transect across a GOES ABI pixel footprint. We will also add a color bar to the figure.

  > This is now Figure 9, and has been moved to the Discussion section. It has been replaced with a larger transect on the scale of the multi-kilometer GOES-R ABI pixel footprints, and a color bar added.

- **Line 53 (L53):** Clarify or define "ABI" on Line 53 to ensure that readers, especially those unfamiliar with the term, can easily follow the content.

  | |
  |---|
  | Advanced Baseline Imager (ABI) has been defined here as it's the first instance that we use the abbreviation. |
  | This has been fixed. |

- **Time Representation:** Consider using local time for a more explicit representation of the expected situation at that time.

  | |
  |---|
  | We will make all of our times local (UTC-7) for consistency and easy of interpretation for the reader. |
  | All times are now reported in local time, UTC-7. |

- **Snow Pit Locations:** Provide information about the locations of the manual snow pits, noting that some may be situated in the shadows of trees. This would be interesting to see the ~10 degree difference of Figure 2 between shadow and illuminated snow.

  | |
  |---|
  | This is another great idea, thank you! We can add this information to the paper to help interpret the result from those few snow pit locations we did compare with the thermal infrared imagery. We may try to actually change figure 2 to a long (GOES ABI pixel scale) transect that passes through one of our snow pit locations. |
  | We now mention the distance each snow pit was from the nearest tree stand (> 50 m in each case) at P11, L273. |

- **Figure 6:** The temperature gradient that appears in the acquisition as reported in Figure 6 is not clear. Line 316 offers an explanation, but it is currently too vague. Please provide a more detailed and explicit clarification to help readers understand the observed temperature gradient more effectively.

  | |
  |---|
  | We will re-word this to clarify and better explain the patterns we saw in the airborne imagery. We may also add to Figure 6 to help illustrate our description. |
  | The discussion around the systematic differences seen between the airborne IR and ASTER brightness temperature images has been expanded upon in Section 5.1. |